# Differential analysis of mean blood glucose levels from venous and fingertip in predicting 30-day mortality among ICU patients with severe trauma: A retrospective study utilizing the MIMIC-IV database

**Fei Yin[1], Zhenguo Qiao[2], Xiaofei Wu[1], Yuzhou Xu[3], Yun Liu[1]***

**1** Department of Emergency, Suzhou Ninth People's Hospital, Suzhou Ninth Hospital Affiliated to Soochow University, Suzhou, Jiangsu, China, **2** Department of Gastroenterology, Suzhou Ninth People's Hospital, Suzhou Ninth Hospital Affiliated to Soochow University, Suzhou, Jiangsu, China, **3** Department of Rehabilitation, Suzhou Ninth People's Hospital, Suzhou Ninth Hospital Affiliated to Soochow University, Suzhou, Jiangsu, China

* liuyun1000@126.com

## Abstract

### Background

To investigate the correlation between mean blood glucose of venous (VMBG) and mean blood glucose of fingertip (FMBG) within 30 days and 30-day mortality in trauma patients in intensive care unit (ICU), and to systematically evaluate the prognostic value of early-stage and long-term monitoring intervals.

### Materials and methods

Utilizing data from the MIMIC-IV database, we employed receiver operating characteristic (ROC) curves, restricted cubic splines (RCS), and Cox proportional hazards models to assess glucose-outcome relationships. Sensitivity analyses using complete datasets, propensity score matching (PSM), and subgroup analyses were conducted to verify result robustness. Furthermore, to minimize the bias of immortal time, the Landmark analysis was employed to investigate the correlation between the early-stage VMBG, FMBG and 30-day mortality rate. Secondary outcomes included 90-, 180-, and 360-day all-cause mortality.

### Results

A total of 2,699 patients were enrolled in the study. The AUC values (95% CI) for VMBG and FMBG were 0.705 (0.675-0.735) and 0.640 (0.608-0.672), respectively. VMBG demonstrated superior predictive ability for 30-day mortality compared to FMBG (Z=5.833, P<0.001). Multivariate-adjusted Cox regression revealed

**Data availability statement:** The MIMIC-IV database were publicly available on the physionet platform (https://physionet.org/content/mimiciv/3.0/).

**Funding:** The project was supported by the 2024 Excellence Program for Young and Middle-aged Talents at Suzhou Ninth People's Hospital. Award Number: Document No. [2024] 23 of Suzhou Ninth People's Hospital Party Committee. We confirm that the funders had no role in study design, data collection and analysis, decision to publish, or preparation of the manuscript.

**Competing interests:** The authors have declared that no competing interests exist.

independent associations between VMBG, FMBG, and 30-day mortality, with HRs of 1.019 (1.016-1.023) and 1.009 (1.006-1.013), respectively. RCS analysis further indicated a nonlinear "J-shaped" relationship between VMBG, FMBG, and outcomes (P for nonlinearity<0.001), with two thresholds at 88.1 mg/dL and 125.4 mg/dL for VMBG, and 95.4 mg/dL and 134.0 mg/dL for FMBG. The threshold values for FMBG were observed to be higher than those of VMBG. According to the thresholds of VMBG and FMBG, patients were stratified into hypoglycemic, normoglycemic, and hyperglycemic groups, respectively. Whether in the pre-PSM or post-PSM cohort, with the normoglycemic group as the reference, both the hypoglycemic and hyperglycemic groups of VMBG and FMBG were associated with an increased 30-day mortality rate. Subgroup analyses revealed that the impact of elevated VMBG and FMBG on prognosis was more pronounced in patients aged <65 years, non-White individuals, and those without diabetes(P for interaction<0.05). VMBG and FMBG during different time intervals were all non - linearly correlated with the 30 - day mortality rate. The Landmark analysis indicated that during the early-stages, there was also a significant statistical correlation among VMBG, FMBG and prognosis. For secondary outcomes, VMBG and FMBG also showed significant associations with 90-day, 180-day, and 360-day all-cause mortality.

## Conclusion

In ICU trauma patients, both VMBG and FMBG across various time intervals exhibited nonlinear associations with 30-day mortality. Although venous blood glucose monitoring typically demonstrated higher prognostic predictive accuracy compared to fingertip measurements, the early - warning value of fingertip blood glucose should not be overlooked. In clinical monitoring, the characteristics of both measurement methods should be comprehensively recognized.

## Introduction

In critically ill patients, hyperglycemia had been strongly associated with a range of adverse outcomes, including increased risk of infections, multi-organ dysfunction, and higher mortality rates [1]. As early as 2001, Van den Berghe et al. introduced the concept of intensive insulin therapy (IIT), aiming to maintain blood glucose levels within a strict range of 80–110 mg/dL. This approach demonstrated the potential to reduce morbidity and mortality among critically ill patients in the surgical intensive care unit (SICU) [2]. Subsequently, multiple prospective studies further supported the efficacy of this treatment strategy [3,4]. However, the findings of the NICE-SUGAR study in 2009—a large-scale, multicenter, international randomized controlled trial— challenged this paradigm. The study revealed that tight glycemic control, compared to maintaining blood glucose levels below 180 mg/dL, did not show significant benefits for critically ill patients and might even be harmful due to an increased incidence

of hypoglycemic events [5]. These conflicting results had sparked ongoing debate within the medical community regarding the optimal glycemic control strategies for critically ill patients.

Following trauma, glucose served as a critical energy substrate essential for maintaining immune cell function, promoting tissue repair, and supporting the cardiovascular system [6]. Trauma-induced hyperglycemia was typically attributed to elevated levels of epinephrine, glucagon, cortisol, and cytokines, which accelerated glycogenolysis and induce peripheral insulin resistance [7]. However, studies showed that both early and sustained hyperglycemia were significantly associated with adverse outcomes in trauma patients, including increased risk of infections, prolonged hospital stays, higher complication rates, and elevated mortality [8–10]. Fakhry et al. further demonstrated that hyperglycemia (>180 mg/dL) was linked to increased mortality risk in both diabetic and non-diabetic trauma patients, with non-diabetic patients experiencing worse outcomes [11]. Additionally, trauma patients with hyperglycemia upon admission faced almost twice the risk of developing coagulopathy [12]. Bosarge et al. also highlighted that stress hyperglycemia was associated with higher mortality rates in patients with severe traumatic brain injury [13]. Given the significant adverse impact of hyperglycemia on trauma patient outcomes, the development of a systematic in-hospital glycemic management protocol was urgently needed. However, current research on optimal glycemic control targets for trauma patients remained limited, with even less attention given to the risks of hypoglycemia. Moreover, most studies failed to adequately account for the heterogeneity introduced by different glucose measurement methods. In fact, under critical conditions, significant discrepancies arose between glucose values obtained through different measurement techniques [14]. To address these gaps, we conducted a retrospective analysis employing the MIMIC database. Leveraging the comprehensive nature of this dataset, we explored the correlation between glucose levels and trauma patient outcomes from two distinct perspectives: venous blood glucose and fingertip blood glucose. Our findings aimed to provide a scientific basis for establishing optimal in-hospital glycemic control targets for trauma patients.

## Method

### Database

This study involved a retrospective observational cohort analysis utilizing data extracted from the MIMIC-IV v3.0 database, which included 94,458 ICU admissions from 2008 to 2022 [15]. These patients were admitted to either the emergency department or ICU of Beth Israel Deaconess Medical Center in Boston, Massachusetts. To gain access to the database, we successfully completed the CITI Program courses related to Human Research and Data or Specimen Only Research (Record ID: 41,696,976). The study obtained approval from the Institutional Review Board of Beth Israel Deaconess Medical Center, which waived the requirement for ethical review and informed consent.

### Patients

We utilized Navicat Premium software (version 16.1.11) to execute Structured Query Language (SQL) queries for extracting data from the MIMIC-IV database. Patients with a primary diagnosis corresponding to trauma codes under ICD-9 (800–959) or ICD-10 (S00–S99, T00–T14, T20–T32) were included in the study [16]. For patients with multiple ICU admissions, data were exclusively collected from their initial admission. The screening criteria were as follows: (1) exclusion of patients younger than 18 years or older than 89 years; (2) exclusion of patients with an ICU stay duration of less than 24 hours; (3) exclusion of patients lacking recorded blood glucose values within 24 hours of admission or with fewer than two blood glucose measurements within 30 days (Fig 1). The recorded variables encompassed demographic information (age, sex, and race), comorbidities, vital signs recorded on the first day of ICU admission, corresponding laboratory test results, Simplified Acute Physiology Score II (SAPS II), Sequential Organ Failure Assessment (SOFA) score, Acute Physiology Score III (APS III), Oxford Acute Severity of Illness Score (OASIS), Glasgow Coma Scale (GCS) score, application of mechanical ventilation, application of continuous renal replacement therapy (CRRT), insulin administration, blood

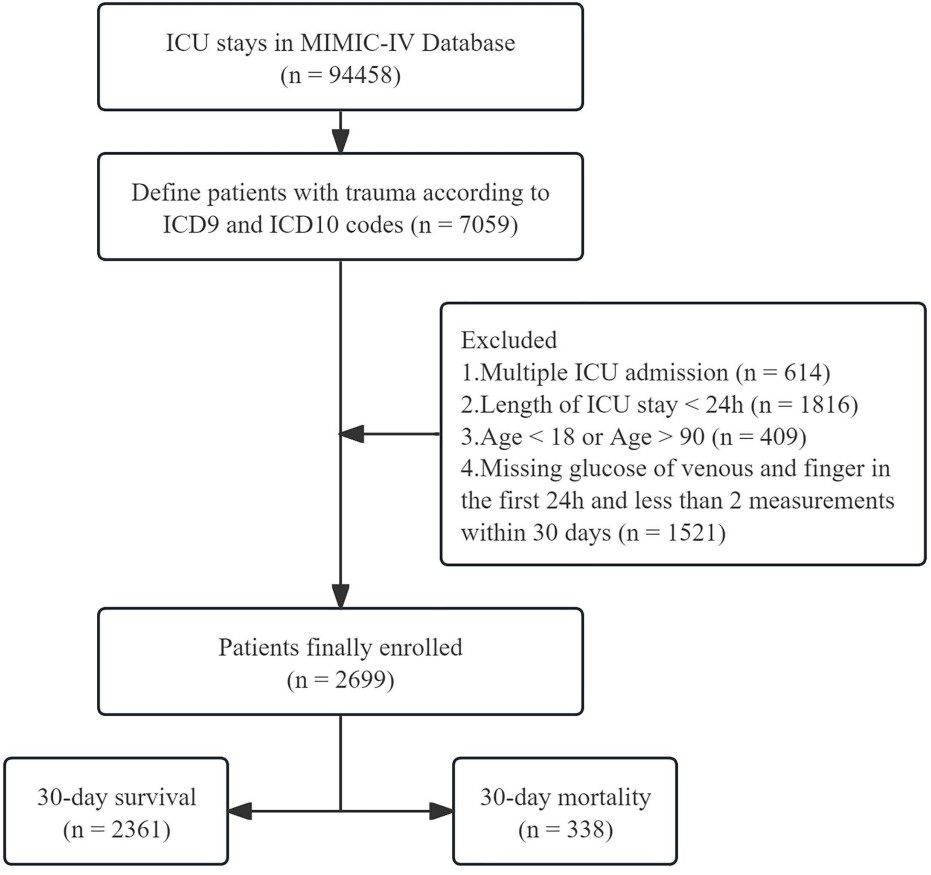

**Fig 1. Patient enrollment selection flowchart.** ICD-9: The ninth revision of the International Classifcation of Diseases. ICD-10: The tenth revision of the International Classifcation of Diseases. ICU: intensive care unit.

transfusion therapy, and maximum acute kidney injury (AKI) stage. We calculated the mean blood glucose of venous (VMBG), the mean blood glucose of fingertip (FMBG), glycemic variability of venous (VGV), and glycemic variability of fingertip (FGV) across various time intervals. All the glucose values utilized in this study were extracted from the MIMIC-IV database, reflecting the actual clinical records. The timing, frequency and method of measurement were not standardized in advance by the research protocol. Glycemic variability (GV) was defined as the ratio of the standard deviation (SD) of blood glucose levels to the mean blood glucose concentration. If a variable had multiple measurements, the average value was used. The primary outcome was defined as 30-day all-cause mortality, while secondary outcomes encompassed all-cause mortality at 90 days, 180 days, and 360 days.

## Statistical analysis

Statistical analyses were conducted using IBM SPSS Statistics (Version 25.0) and RStudio software (Version 2022.07.0). Normally distributed continuous variables were presented as mean ± standard deviation (SD) and analyzed using the independent samples t-test, while non-normally distributed continuous variables were expressed as median and interquartile range (IQR) and compared using the Mann-Whitney U test. Categorical variables were summarized as frequencies and percentages and assessed using the chi-square test. A two-tailed p-value < 0.05

was considered to indicate statistical significance. Patients were stratified into survival and mortality groups according to their 30-day survival outcomes. Missing data were handled using the random forest multiple imputation technique, and extreme values were handled by applying a 1% tail trimming approach [15]. Receiver operating characteristic (ROC) curves were plotted to evaluate the predictive performance of VMBG, FMBG, VGV, and FGV. Subsequently, the areas under the curves (AUCs) were compared to provide a quantitative evaluation of their respective predictive capabilities. There existed an inherent mathematical connection between MBG and GV. Owing to this methodological limitation, it was not incorporated into the main multivariate model to prevent interpretational difficulties. The nonlinear associations between VMBG, FMBG, and 30-day mortality were investigated using five-knot restricted cubic splines (RCS). Patients were classified into three categories according to the RCS findings. Cox proportional hazards regression models were utilized to evaluate the relationships between VMBG, FMBG, and mortality at 30-day, 90-day, 180-day, and 360-day intervals in trauma patients. To optimize the balance–precision trade-off, propensity score matching (PSM) analysis was performed using a nearest-neighbor matching algorithm with a caliper width of 0.4 at a 2:1 ratio, and a sensitivity analysis was conducted using caliper width of 0.2. Standardized mean differences (SMDs) were calculated to evaluate the balance between groups, and the associations between VMBG, FMBG, and 30-day mortality were reanalyzed after PSM. To minimize the bias of immortal time, the associations between VMBG, FMBG, and 30-day mortality rates across different time intervals (including the early-stage) were investigated. Sensitivity analysis was conducted using the complete dataset. Finally, forest plots were generated to present subgroup analysis results and assess potential interaction effects. Through the application of the variance inflation factor (VIF) test, it was determined that no multicollinearity existed among all the covariates in the multivariate models.

## Result

### Baseline characteristics

A total of 2,699 eligible patients were enrolled in this study and stratified into two distinct cohorts based on 30-day outcomes: the mortality group (n = 338) and the survival group (n = 2,361). The number of venous blood glucose measurements was 9[5,17], and 2337 cases (86.6%) had four or more measurements. The number of fingertip blood glucose measurements was 7[4,17], and 2154 patients (79.8%) had four or more measurements. The distribution of glucose measurement numbers for all patients was presented in S1 Fig. The mortality group demonstrated a significantly prolonged ICU length of stay (median 4.64 days vs. survival group 2.98 days, p < 0.001), while exhibiting a 21.3% shorter overall hospitalization duration (median 7.16 days vs. 9.07 days, p < 0.001) when compared with the survival cohort. The comorbidity index was significantly elevated in the mortality group, demonstrating a higher prevalence of conditions such as cerebrovascular disease, congestive heart failure, liver disease, renal disease, cancer, and diabetes compared to the survival group. The survival cohort demonstrated significantly younger age (median 63 vs. 77 years) and superior clinical metrics across multiple domains compared with the mortality group. Notable differences included: 1) Disease severity scores: lower SOFA, SAPS II, APS III, OASIS; 2) Vital signs: lower respiratory rate, percutaneous oxygen saturation (SpO2); 3) Glucose metabolism: reduced VMBG, FMBG, VGV, FGV; 4) Laboratory values: lower red cell distribution width (RDW), white blood cell (WBC) count, creatinine, blood urea nitrogen (BUN), sodium, chloride, anion gap (AG), prothrombin time (PT), partial thromboplastin time (PTT), international normalized ratio (INR); 5) Therapeutic interventions: lower utilization rates of mechanical ventilation, blood transfusion, CRRT and insulin therapy. Conversely, the mortality group exhibited significantly lower values in diastolic blood pressure (DBP), mean arterial pressure (MBP), red blood cell (RBC) count, hemoglobin, hematocrit, platelet count, and bicarbonate levels than the survival group. The GCS score did not differ significantly between the groups. Additionally, statistically significant disparities were observed in ethnicity, sex distribution, and AKI staging across the two groups (Table 1).

**Table 1. Baseline demographic and clinical characteristics of participants.**

| Variables | Overall | 30-day survial | 30-day mortality | t/z/χ2 | p |
|---|---|---|---|---|---|
| N | 2699 | 2361 | 338 | | |
| Age (year) | 64.71 [48.78, 78.42] | 63.23 [46.93, 76.57] | 77.30 [63.25, 84.41] | −10.717 | <0.001 |
| Male (%) | 1782 (66.0) | 1583 (67.0) | 199 (58.9) | −2.966 | 0.004 |
| Weight (kg) | 77.40 [65.47, 91.63] | 78.00 [66.00, 92.10] | 73.22 [61.70, 87.08] | −4.054 | <0.001 |
| Race (%) | | | | 31.074 | <0.001 |
| Other | 747 (27.7) | 627 (26.6) | 120 (35.5) | | |
| Hispanic | 117 (4.3) | 114 (4.8) | 3 (0.9) | | |
| Black | 143 (5.3) | 138 (5.8) | 5 (1.5) | | |
| Asian | 54 (2.0) | 50 (2.1) | 4 (1.2) | | |
| White | 1638 (60.7) | 1432 (60.7) | 206 (60.9) | | |
| *Vital signs* | | | | | |
| VMBG (mg/dL) | 125.14 [110.84, 147.47] | 122.56 [109.50, 142.83] | 146.56 [127.62, 175.77] | −12.233 | <0.001 |
| FMBG (mg/dL) | 133.60 [116.89, 156.40] | 131.82 [115.50, 153.83] | 150.66 [129.50, 174.64] | −8.361 | <0.001 |
| VGV (%) | 17.65[12.82, 25.16] | 17.27[12.64, 24.20] | 20.13[14.47, 29.97] | −5.272 | <0.001 |
| FGV (%) | 16.79[11.76, 23.34] | 16.55[11.47, 22.77] | 20.20[13.79, 29.01] | −6.258 | <0.001 |
| Heart rate (bpm) | 83.41 [72.53, 95.73] | 83.50 [72.52, 95.68] | 82.16 [72.56, 96.15] | −0.005 | 0.996 |
| SBP (mmHg) | 123.05 [112.40, 133.44] | 123.05 [112.42, 133.43] | 122.87 [112.31, 133.46] | −0.303 | 0.762 |
| DBP (mmHg) | 64.07 [57.65, 72.07] | 64.56 [58.19, 72.39] | 60.98 [54.85, 68.55] | −6.000 | <0.001 |
| MBP (mmHg) | 80.53 [74.07, 87.73] | 80.74 [74.11, 88.17] | 79.36 [73.35, 85.67] | −2.626 | 0.009 |
| Resp rate (bpm) | 17.96 [16.22, 20.12] | 17.85 [16.13, 19.96] | 18.75 [17.00, 21.68] | −6.239 | <0.001 |
| Temperature (℃) | 37.00 [36.75, 37.31] | 37.00 [36.76, 37.30] | 36.97 [36.64, 37.37] | −1.266 | 0.276 |
| Spo2 (%) | 97.73 [96.28, 99.02] | 97.64 [96.24, 98.92] | 98.44 [96.73, 99.48] | −4.854 | <0.001 |
| *Scoring systems* | | | | | |
| GCS | 14.00 [13.00, 15.00] | 14.00 [13.00, 15.00] | 14.50 [11.00, 15.00] | −1.991 | 0.046 |
| SOFA | 3.00 [2.00, 5.00] | 3.00 [2.00, 5.00] | 5.00 [3.00, 8.00] | −12.347 | <0.001 |
| SAPSII | 32.00 [24.00, 40.00] | 30.00 [23.00, 38.00] | 42.50 [35.00, 50.00] | −16.706 | <0.001 |
| APSIII | 36.00 [28.00, 48.00] | 35.00 [27.00, 45.00] | 49.00 [37.00, 61.00] | −12.881 | <0.001 |
| OASIS | 31.00 [26.00, 37.00] | 31.00 [26.00, 36.00] | 36.50 [32.00, 42.00] | −12.679 | <0.001 |
| AKI Stage (%) | | | | 80.895 | <0.001 |
| 0 | 679 (25.2) | 634 (26.9) | 45 (13.3) | | |
| 1 | 502 (18.6) | 453 (19.2) | 49 (14.5) | | |
| 2 | 1110 (41.1) | 967 (41.0) | 143 (42.3) | | |
| 3 | 408 (15.1) | 307 (13.0) | 101 (29.9) | | |
| *Laboratory parameters* | | | | | |
| Hematocrit (%) | 33.90 [29.56, 37.88] | 34.13 [29.90, 38.03] | 31.80 [28.01, 36.24] | −5.642 | <0.001 |
| Hemoglobin (g/dL) | 11.43 [9.82, 12.76] | 11.55 [10.00, 12.85] | 10.59 [9.13, 11.86] | −7.019 | <0.001 |
| Platelets (10^9/L) | 194.00 [149.88, 241.00] | 197.00 [152.67, 243.00] | 174.08 [131.25, 220.94] | −5.175 | <0.001 |
| WBC (10^9/L) | 11.25 [8.63, 14.31] | 11.17 [8.60, 14.15] | 11.95 [8.94, 15.46] | −2.787 | 0.005 |
| RBC (10^12/L) | 3.63 [3.15, 4.10] | 3.66 [3.18, 4.12] | 3.42 [2.97, 3.85] | −5.664 | <0.001 |
| RDW (%) | 13.90 [13.17, 15.00] | 13.80 [13.10, 14.85] | 14.60 [13.60, 15.99] | −8.208 | <0.001 |
| Anion gap (mmol/L) | 14.00 [12.00, 16.00] | 14.00 [12.00, 16.00] | 15.00 [12.75, 17.00] | −4.622 | <0.001 |
| Bicarbonate (mmol/L) | 23.25 [21.00, 25.42] | 23.50 [21.25, 25.50] | 22.00 [19.50, 24.50] | −5.873 | <0.001 |
| Bun (mg/dL) | 16.00 [11.60, 22.00] | 15.33 [11.25, 21.00] | 19.71 [15.00, 30.00] | −9.210 | <0.001 |
| Calcium (mmol/L) | 8.37 [7.90, 8.80] | 8.35 [7.90, 8.80] | 8.40 [7.97, 8.85] | −1.380 | 0.155 |
| Chloride (mmol/L) | 104.50 [101.50, 107.50] | 104.50 [101.50, 107.50] | 105.00 [101.50, 109.00] | −2.642 | 0.008 |
| Creatinine (mg/dL) | 0.90 [0.70, 1.11] | 0.87 [0.70, 1.10] | 1.00 [0.78, 1.43] | −6.385 | <0.001 |

*(Continued)*

**Table 1.** (Continued)

| Variables | Overall | 30-day survival | 30-day mortality | t/z/χ2 | p |
|---|---|---|---|---|---|
| Sodium (mmol/L) | 139.00 [137.00, 141.00] | 139.00 [137.00, 141.00] | 139.50 [137.27, 142.48] | −3.795 | <0.001 |
| Potassium (mmol/L) | 4.10 [3.80, 4.45] | 4.10 [3.80, 4.45] | 4.16 [3.81, 4.50] | −1.408 | 0.156 |
| INR | 1.15 [1.07, 1.30] | 1.13 [1.05, 1.25] | 1.25 [1.10, 1.44] | −8.511 | <0.001 |
| PT | 12.80 [11.73, 14.20] | 12.70 [11.70, 14.00] | 13.70 [12.31, 15.69] | −7.930 | <0.001 |
| PTT | 27.50 [25.20, 30.65] | 27.30 [25.12, 30.40] | 28.83 [26.10, 33.08] | −6.177 | <0.001 |
| *Comorbidities* | | | | | |
| Comorbidity index | 3.00 [1.00, 5.00] | 3.00 [1.00, 5.00] | 5.00 [3.00, 7.00] | 148.783 | <0.001 |
| Congestive heart failure (%) | 350 (13.0) | 280 (11.9) | 70 (20.7) | 20.522 | <0.001 |
| Cerebrovascular disease (%) | 247 (9.2) | 195 (8.3) | 52 (15.4) | 18.056 | <0.001 |
| Chronic pulmonary disease (%) | 427 (15.8) | 364 (15.4) | 63 (18.6) | 2.305 | 0.15 |
| Renal disease (%) | 313 (11.6) | 246 (10.4) | 67 (19.8) | 25.501 | <0.001 |
| Liver disease (%) | 185 (6.9) | 143 (6.1) | 42 (12.4) | 12.787 | <0.001 |
| Cancer (%) | 99 (3.7) | 75 (3.2) | 24 (7.1) | 12.884 | 0.001 |
| Diabetes (%) | 719 (26.6) | 602 (25.5) | 117 (34.6) | 12.577 | 0.001 |
| *Treatment* | | | | | |
| CRRT (%) | 28 (1.0) | 11 (0.5) | 17 (5.0) | 59.981 | <0.001 |
| Ventilation (%) | 1451 (53.8) | 1190 (50.4) | 261 (77.2) | 85.534 | <0.001 |
| Transfusion (%) | 947 (35.1) | 811 (34.3) | 136 (40.2) | 4.499 | 0.039 |
| Insulin (%) | 1401 (51.9) | 1185 (50.2) | 216 (63.9) | 22.278 | <0.001 |
| *Outcomes* | | | | | |
| Los hospital (days) | 8.83 [5.32, 15.96] | 9.07 [5.57, 16.64] | 7.16 [3.94, 11.70] | −43.643 | <0.001 |
| Los icu (days) | 3.13 [1.85, 6.48] | 2.98 [1.82, 5.95] | 4.64 [2.53, 8.46] | −6.579 | <0.001 |
| In-hospital mortality (%) | 274 (10.2) | 13 (0.6) | 261 (77.2) | 1905.396 | <0.001 |
| 90-day mortality (%) | 433 (16.0) | 95 (4.0) | 338 (100.0) | 2022.067 | <0.001 |
| 180-day mortality (%) | 507 (18.8) | 169 (7.2) | 338 (100.0) | 1670.537 | <0.001 |
| 360-day mortality (%) | 585 (21.7) | 247 (10.5) | 338 (100.0) | 1396.281 | <0.001 |

VMBG: mean blood glucose of venous within 30 days. FMBG: mean blood glucose of fingertip within 30 days. VGV: glycemic variability of venous within 30 days. FGV: glycemic variability of fingertip within 30 days. DBP: diastolic blood pressure. SBP: systolic blood pressure. MAP: mean arterial pressure. RDW: red cell distribution width. RBC: red blood cell. WBC: white blood cell. PT: prothrombin time. INR: international normalized ratio. PTT: partial thromboplastin time. SAPS II: Simplified Acute Physiology Scores II. APS III: Acute Physiology Score III. SOFA: Sequential Organ Failure Assessment. GCS: Glasgow Coma Scale. OASIS: Oxford Acute Severity of Illness Score. AKI stage: acute kidney injury stage. CRRT: continuous renal replacement therapy.

## ROC curve analysis

The prognostic efficacy of VMBG, FMBG, VGV, FGV for 30-day mortality was compared via ROC curve analysis in critically ill trauma patients requiring intensive care. The AUC (95%CI) values were 0.705 (0.675–0.735) for VMBG, 0.640 (0.608–0.672) for FMBG, 0.589 (0.556–0.622) for VGV, and 0.605 (0.572–0.638) for FGV (Fig 2). Notably, VMBG demonstrated significantly superior discriminative capacity relative to FMBG (Z = 5.833, P < 0.001), VGV (Z = 5.966, P < 0.001), and FGV (Z = 4.926, P < 0.001). Although FMBG showed better predictive performance than VGV (Z = 2.470, P = 0.014), its predictive advantage over FGV failed to achieve statistical significance (Z = 1.671, P = 0.095).

## Cox regression and restricted cubic spline (RCS) analysis

To investigate the potential independent association between VMBG, FMBG and 30-day mortality in critically injured trauma patients, we systematically conducted univariable and multivariable Cox proportional hazards regression analyses

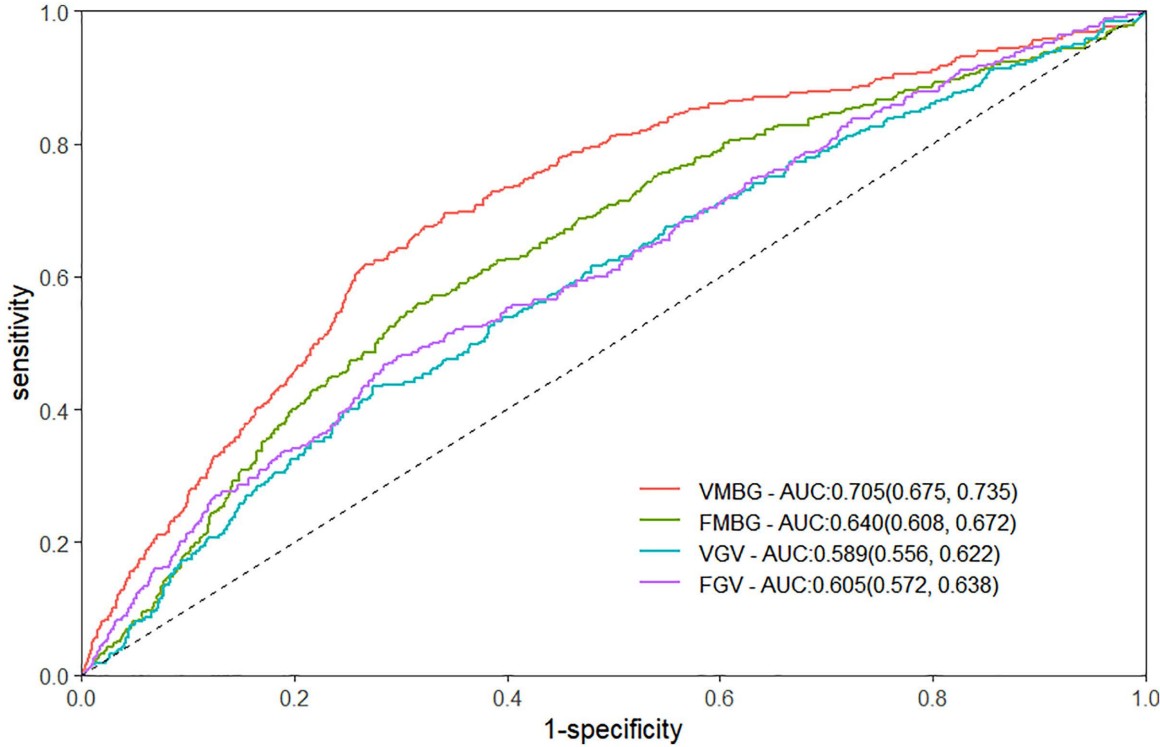

**Fig 2. ROC curves of the VMBG, FMBG, VGV, FGV to predict 30-day mortality among trauma patients.** VMBG: mean blood glucose of venous. FMBG: mean blood glucose of fingertip. VGV: glycemic variability of venous. FGV: glycemic variability of fingertip. ROC: receiver operating characteristic. AUC: area under the curve.

(Table 2). In the unadjusted crude model, both VMBG and FMBG were significantly associated with an increased mortality risk (VMBG: HR = 1.017, 95% CI: 1.014–1.020; FMBG: HR = 1.011, 95% CI: 1.008–1.014). Following comprehensive multivariable adjustment for demographic and clinical confounders (Model 3), the magnitude of association remained statistically significant, revealing 1.019-fold (HR = 1.019, 95% CI 1.015–1.023) and 1.009-fold (HR = 1.009, 95% CI 1.005–1.013) increased in mortality risk per unit glucose increment for VMBG and FMBG respectively. The C-index of the VMBG and FMBG multivariate regression models was 0.833, 0.816, and the Brier Score was 0.083, 0.087.

We conducted a 5-knot RCS analysis in the fully adjusted Model 3 to characterize the non-linear association between VMBG, FMBG, and 30-day mortality in critically injured trauma patients. The results demonstrated a J-shaped nonlinear association for both VMBG and FMBG with 30-day mortality (P for nonlinear <0.001 and 0.004, respectively). Quantitative analysis identified critical glycemic thresholds at 88.1 mg/dL (lower) and 125.4 mg/dL (upper) for VMBG, with corresponding FMBG thresholds at 95.4 mg/dL (lower) and 134.0 mg/dL (upper) (Fig 3). Using these biologically significant cutoff values, the cohort was stratified into three distinct glycemic groups: hypoglycemic (<88.1 mg/dL), normoglycemic (88.1–125.4 mg/dL), and hyperglycemic (>125.4 mg/dL) ranges for VMBG analysis. Similarly, patients were stratified into hypoglycemic (<95.4 mg/dL), normoglycemic (95.4–134.0 mg/dL), and hyperglycemic (>134.0 mg/dL) ranges for FMBG analysis. A piecewise multivariable Cox proportional hazards model was applied to analyze risk differences among the predefined glycemic strata. Compared to the normoglycemic group of VMBG (88.1–125.4 mg/dL), both the hypoglycemic (<88.1 mg/dL) and hyperglycemic (>125.4 mg/dL) groups of VMBG showed significantly higher 30-day mortality risk, with HRs of 2.800 (95% CI: 1.268–6.182) and 3.128 (95% CI: 2.319–4.218), respectively. Analogously, compared to the normoglycemic group of FMBG (95.4–134.0 mg/dL), the hypoglycemic (<95.4 mg/dL) and hyperglycemic (>134.0 mg/dL)

**Table 2. Cox regression analysis between VMBG, FMBG and 30-day mortality in critically injured trauma patients.**

| Variable | Crude | | Model 1 | | Model 2 | | Model 3 | |
|---|---|---|---|---|---|---|---|---|
| | HR(95%CI) | P | HR(95%CI) | P | HR(95%CI) | P | HR(95%CI) | P |
| **VMBG** | | | | | | | | |
| <88.1 mg/dl | 3.076(1.412,6.701) | 0.005 | 2.508(1.150,5.471) | 0.021 | 2.346(1.072,5.133) | 0.033 | 2.800(1.268,6.182) | 0.011 |
| 88.1–125.4 mg/dl | 1(reference) | – | 1(reference) | – | 1(reference) | – | 1(reference) | – |
| ≥125.4 mg/dl | 4.200(3.212,5.492) | <0.001 | 3.594(2.745,4.706) | <0.001 | 3.963(2.990,5.253) | <0.001 | 3.128(2.319,4.218) | <0.001 |
| Continuous | 1.017(1.014,1.020) | <0.001 | 1.016(1.013,1.019) | <0.001 | 1.021(1.018,1.024) | <0.001 | 1.019(1.016,1.023) | <0.001 |
| **FMBG** | | | | | | | | |
| <95.4 mg/dl | 1.686(1.016,2.798) | 0.043 | 1.886(1.135,3.133) | 0.014 | 1.780(1.070,2.961) | 0.026 | 1.905(1.136,3.196) | 0.015 |
| 95.4–134.0 mg/dl | 1(reference) | – | 1(reference) | – | 1(reference) | – | 1(reference) | – |
| ≥134.0 mg/dl | 2.525(1.977,3.224) | <0.001 | 2.170(1.695,2.777) | <0.001 | 2.212(1.704,2.871) | <0.001 | 1.910(1.440,2.535) | <0.001 |
| Continuous | 1.011(1.008,1.014) | <0.001 | 1.009(1.006,1.012) | <0.001 | 1.010(1.007,1.014) | <0.001 | 1.009(1.006,1.013) | <0.001 |

VMBG: mean blood glucose of venous. FMBG: mean blood glucose of fingertip. Crude: no covariates were adjusted. Model1: adjusted for sex, age, race. Model2: adjusted for sex, age, race, comorbidity index, cerebrovascular disease, liver disease, chronic pulmonary disease, diabetes, congestive heart failure, cancer, renal disease. Model3: adjusted for sex, age, race, comorbidity index, cerebrovascular disease, liver disease, chronic pulmonary disease, diabetes, congestive heart failure, cancer, renal disease, CRRT, ventilation, insulin, transfusion, SOFA, GCS, AKI stage, SAPSII, APSIII, OASIS.

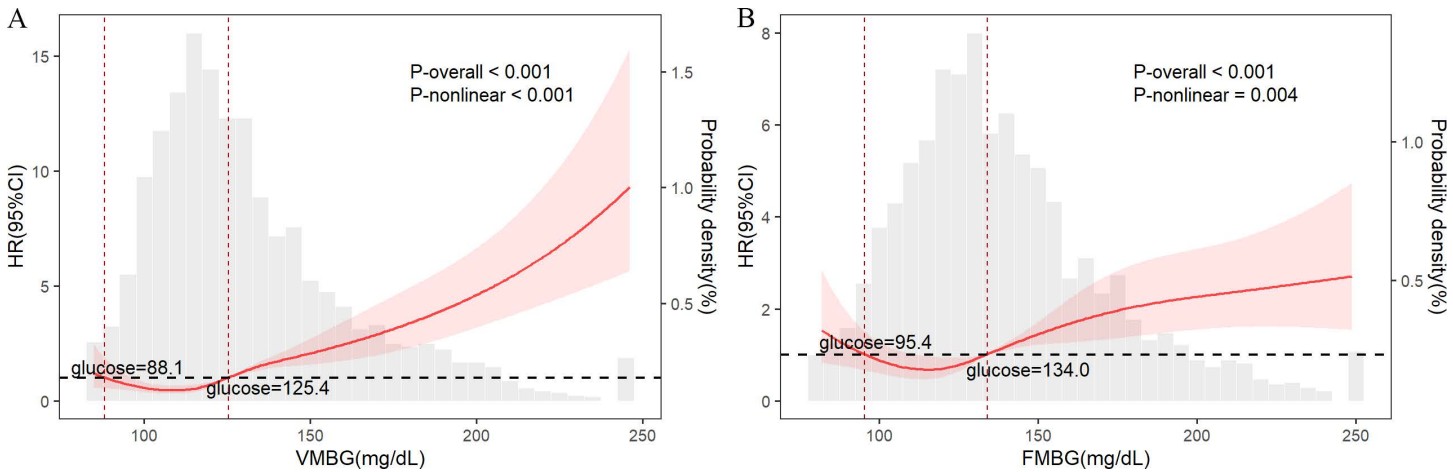

**Fig 3. RCS plot for the association between VMBG and 30-day all-cause mortality (A).** RCS plot for the association between FMBG and 30-day all-cause mortality (B). RCS: restricted cubic spline. VMBG: mean blood glucose of venous. FMBG: mean blood glucose of fingertip. Adjusted for sex, age, race, comorbidity index, cerebrovascular disease, liver disease, chronic pulmonary disease, diabetes, congestive heart failure, cancer, renal disease, CRRT, ventilation, insulin, transfusion, SOFA, GCS, AKI stage, SAPSII, APSIII, OASIS.

groups of FMBG were associated with significantly increased 30-day mortality risk, with HRs of 1.905 (95% CI: 1.136–3.196) and 1.910 (95% CI: 1.440–2.535), respectively.

In secondary outcomes analysis, significant associations persisted between VMBG, FMBG, and long-term all-cause mortality. The adjusted hazard ratios (HRs) for VMBG were 1.017 (95% CI: 1.014–1.020) at 90 days, 1.015 (1.012–1.018) at 180 days, and 1.013 (1.010–1.016) at 360 days. Corresponding HRs for FMBG demonstrated 1.007 (1.004–1.011), 1.006 (1.003–1.009), and 1.005 (1.002–1.008) across the same time periods (Table 3).

**Table 3. Multivariable Cox regression analysis between VMBG, FMBG and all-cause mortality at 90 days, 180 days and 360 days in trauma patients.**

| Variable | 90-day mortality | | 180-day mortality | | 360-day mortality | |
|---|---|---|---|---|---|---|
| | HR(95%CI) | P | HR(95%CI) | P | HR(95%CI) | P |
| VMBG | 1.017(1.014,1.020) | <0.001 | 1.015(1.012,1.018) | <0.001 | 1.013(1.010,1.016) | <0.001 |
| FMBG | 1.007(1.004,1.011) | <0.001 | 1.006(1.003,1.009) | <0.001 | 1.005(1.002,1.008) | <0.001 |

VMBG: mean blood glucose of venous. FMBG: mean blood glucose of fingertip. Adjusted for sex, age, race, comorbidity index, cerebrovascular disease, liver disease, chronic pulmonary disease, diabetes, congestive heart failure, cancer, renal disease, CRRT, ventilation, insulin, transfusion, SOFA, GCS, AKI stage, SAPSII, APSIII, OASIS.

## The associations between VMBG, FMBG, and 30-day mortality across various time intervals

Compared to 30-day survivors, deceased patients had consistently higher mean levels of both venous (VMBG) and fingertip (FMBG) blood glucose across all measured intervals (24-hour, 2-, 3-, 5,- 10-, 20-, and 30-day) (S1 Table). The predictive performance of VMBG and FMBG for 30-day mortality in critically ill trauma patients was evaluated using ROC curve analysis at different time intervals. The AUC (95% CI) values for VMBG were 0.616 (0.584–0.648) within 24 hours, 0.634 (0.601–0.666) within 2 days, 0.656 (0.624–0.688) within 3 days, 0.672 (0.641–0.704) within 5 days, 0.695 (0.665–0.726) within 10 days, and 0.703 (0.673–0.733) within 20 days. For FMBG, the AUC (95% CI) values were 0.615 (0.582–0.648) within 24 hours, 0.619 (0.586–0.652) within 2 days, 0.628 (0.596–0.661) within 3 days, 0.629 (0.596–0.662) within 5 days, 0.639 (0.607–0.671) within 10 days, and 0.640 (0.608–0.672) within 20 days. Intergroup comparisons revealed that, except for the non-significant difference between VMBG and FMBG within 24 hours and 2 days, the AUC of VMBG was significantly higher than that of FMBG at all other time intervals (all P < 0.001). Intragroup comparisons showed that the AUC of VMBG within 30 days was significantly elevated compared to other time intervals. In contrast, the AUC of FMBG within 30 days showed significant increases relative to 24-hours, 2-day, 3-day, and 5-days intervals, but did not exhibit comparable values to those obtained at 10-days and 20-days intervals (Fig 4A).

Multivariable-adjusted Cox regression analysis revealed that the associations between VMBG, FMBG, and 30-day prognosis exhibited a trend similar to the area under the ROC curve, with strengthened correlations over extended time intervals (Fig 4B, S2 Table). The RCS analysis disclosed a J-shaped non-linear association between the two blood glucose indicators, VMBG and FMBG, and the 30-day mortality rate across different time intervals. With the exception of the 24-hour VMBG, which had only one threshold, both FMBG and VMBG in other time periods exhibited two thresholds (at HR = 1). As the time intervals extended and more glucose data were included, the values of the two thresholds gradually approached those of the 30-day interval. Furthermore, the thresholds of FMBG were nearly always higher than those of VMBG across all time intervals (S2 and S3 Figs).

## The associations between VMBG, FMBG, and 30-day mortality of the Landmark cohorts

To conduct a more rigorous assessment of the association between early blood glucose levels and long-term mortality risk and effectively mitigate the immortal time bias, this study further performed a Landmark analysis. The Landmark time point was designated as the 3rd day subsequent to the patient's admission. A new analysis cohort was established which encompassed all patients who survived until the third day and had not been discharged (n = 2516, accounting for 93.2% of the original cohort). In this cohort, VMBG and FMBG for each patient were calculated within the initial 3 days following admission to the ICU. The results demonstrated that the VMBG and FMBG of the deceased group were significantly higher than those of the survivors (S3 Table). Subsequently, these two indicators were respectively utilized as exposure variables. Multivariate Cox proportional hazards regression model was applied to evaluate their association with the mortality risk from the Landmark time point (the third day after admission) to the end of the 30th day. The analysis

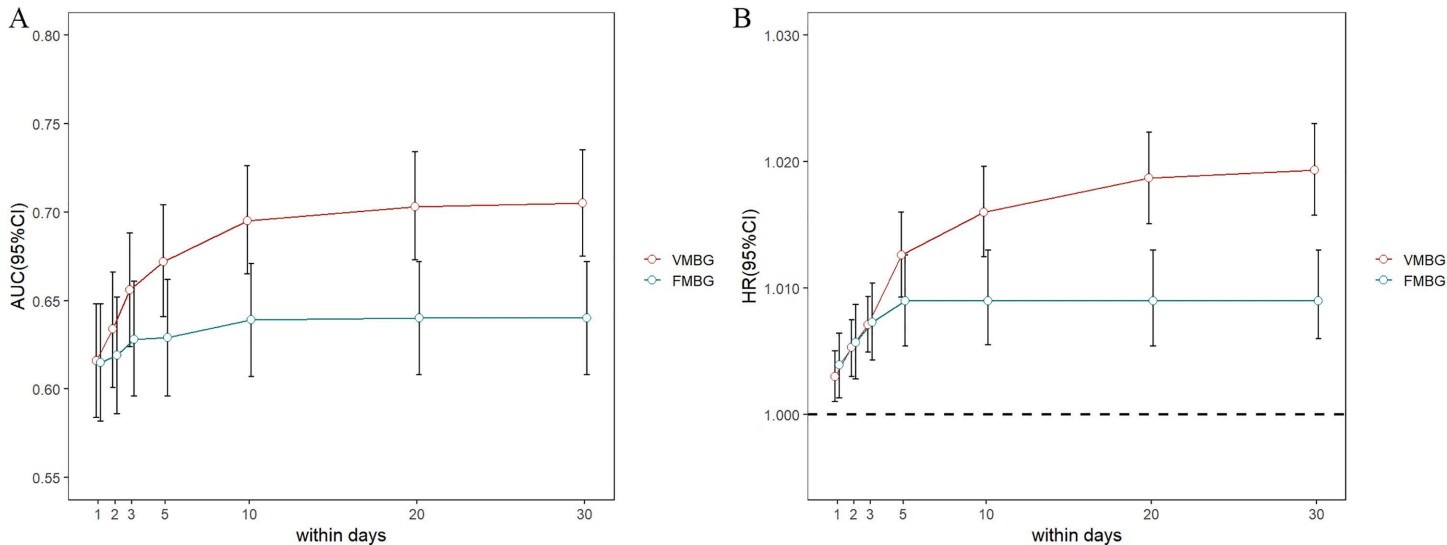

**Fig 4. The AUCs (A) and multivariate adjusted COX regression analysis (B) of VMBG and FMBG in different time intervals post-ICU admission for predicting 30-day all-cause mortality.** Adjusted for sex, age, race, comorbidity index, cerebrovascular disease, liver disease, chronic pulmonary disease, diabetes, congestive heart failure, cancer, renal disease, CRRT, ventilation, insulin, transfusion, SOFA, GCS, AKI stage, SAPSII, APSIII, OASIS. AUC: area under curve.

revealed that both VMBG and FMBG within 3 days were significantly positively correlated with the 30-day mortality risk: the adjusted hazard ratio (HR) for VMBG was 1.006 (95% CI: 1.003–1.010), and the adjusted HR for FMBG was 1.005 (95% CI: 1.001–1.009) (S4 Table). When the sensitivity analysis was conducted at the landmark time point designated as the 2nd day after the patient's admission, the results were also encouraging. Since patients with a hospital stay of less than 24 hours had been excluded, this time period was no longer subject to the Landmark analysis.

### Sensitivity analysis

Sensitivity analyses using the complete dataset (S5 Table, excluding patients with any missing variables) demonstrated that the risk estimates for unadjusted VMBG, FMBG, and progressively adjusted for model 1, model 2 and model 3 (S6 Table) showed no statistically significant discrepancies compared with results from the multiple imputation dataset. These findings validate the robustness of the multiple imputation approach and suggest that the missing data mechanism likely follows missing completely at random (MCAR) patterns, exerting no substantial impact on the study conclusions. We ran the main model on the original data without 1% tail trimming. The results were largely consistent with those of the main analysis, indicating that the conclusions drawn were not sensitive to the treatment of extreme values (S7 and S8 Tables).

### Propensity score matching (PSM) analysis

All baseline covariates were incorporated into the PSM model to optimize covariate balance. Given the relatively small sample size in the low-level subgroups after triage based on VMBG or FMBG in the original cohort, a caliper width of 0.4 standard deviations was implemented with a 2:1 nearest-neighbor matching algorithm to maximize sample retention while maintaining cohort balance. The VMBG and FMBG demonstrated greater comparability within the matched cohort (Fig 5). As detailed in S9 Table and Fig 6, post-PSM analyses of baseline characteristics and SMDs between survival and mortality groups revealed absolute SMDs below 0.1 for all matched covariates, indicating satisfactory balance after matching. This balance was verified through a sensitivity analysis employing a 0.2 caliper width (S4 Fig).

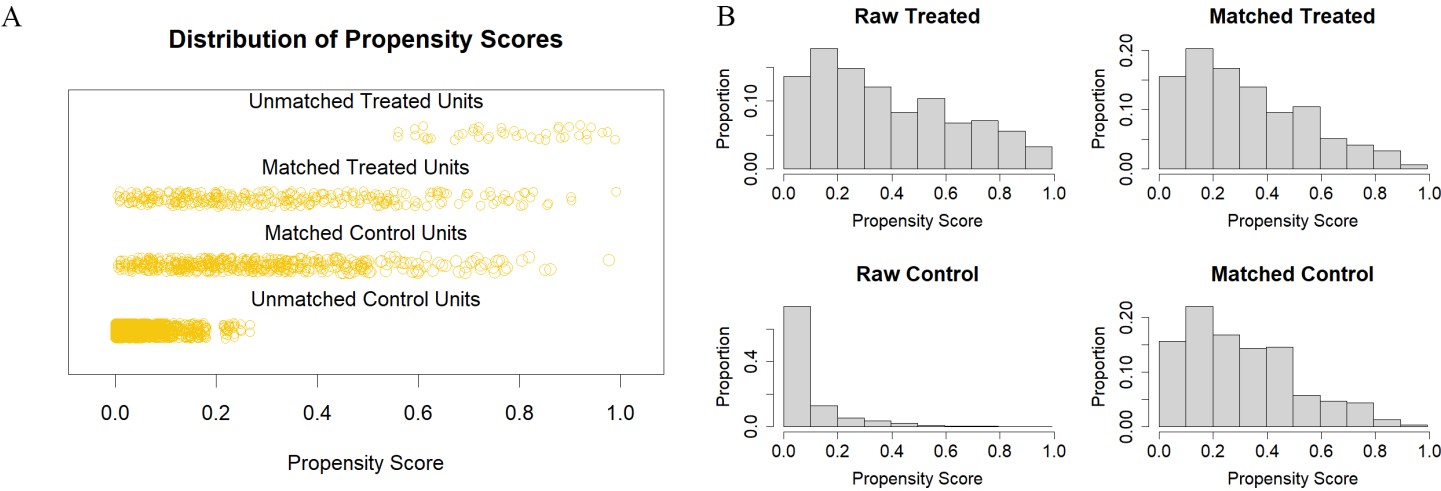

**Fig 5. Jitter plot illustrating the distribution of propensity scores (A).** Histogram representing the distribution of propensity scores (B).

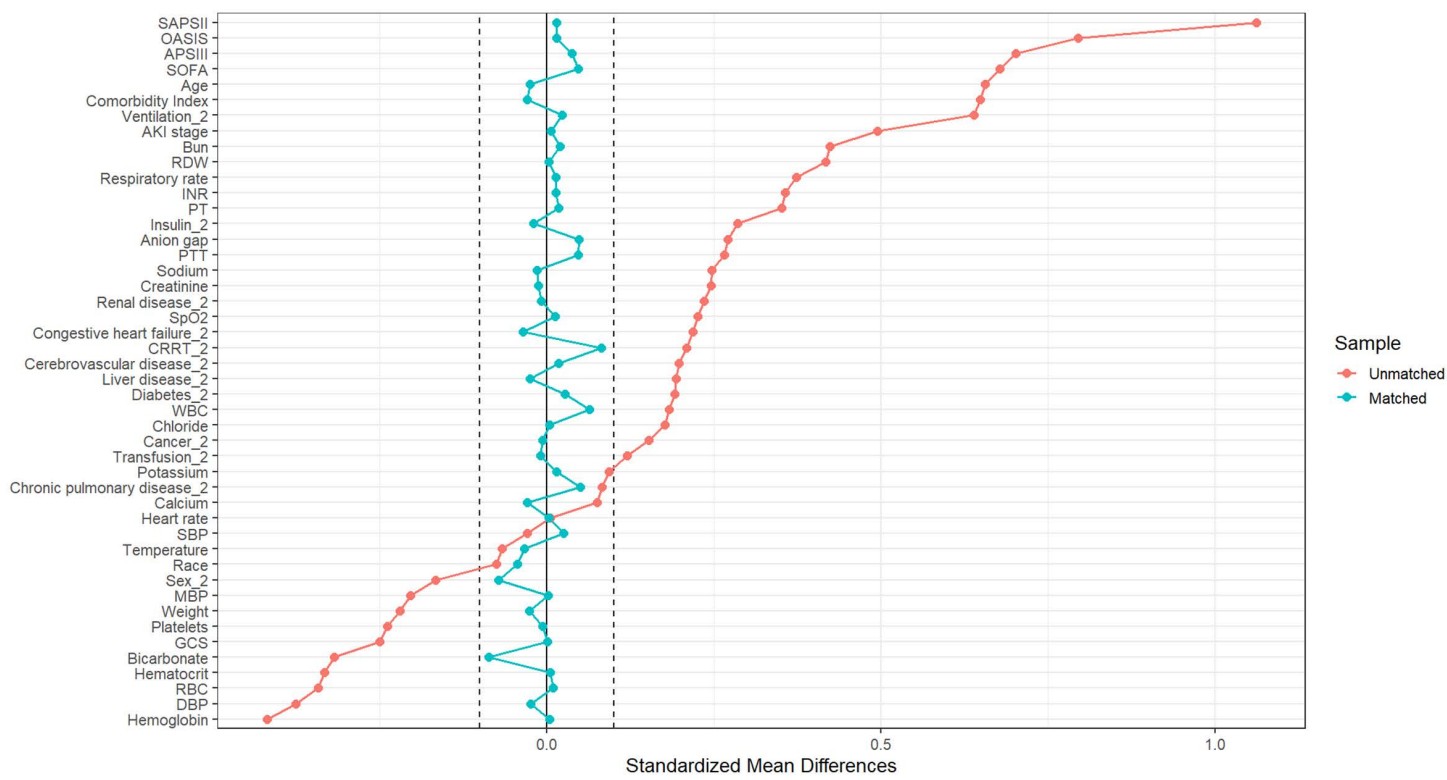

**Fig 6. SMDs of variables prior to and following the matching process.** SMDs: standardized mean differences.

In the post-PSM cohort, Cox proportional hazards regression analyses were conducted to assess the associations between VMBG, FMBG and mortality risk, treating these variables both as continuous and categorical measures. In continuous analyses, each 1 mg/dL increment in VMBG and FMBG was associated with 1.2% and 0.5% increased 30-day

mortality risk, respectively. In the three-category analysis (low, intermediate, high), using the intermediate VMBG category (88.1–125.4 mg/dL) as the reference, the low (<88.1 mg/dL) and high (>125.4 mg/dL) VMBG groups exhibited significantly elevated mortality risks, with adjusted HRs of 2.89 (95% CI: 2.12–3.94) and 2.51 (95% CI: 1.85–3.39), respectively. Similarly, when compared to the intermediate FMBG category (95.4–134.0 mg/dL), the low (<95.4 mg/dL) and high (>134.0 mg/dL) FMBG groups demonstrated HRs of 2.38 (95% CI: 1.76–3.22) and 1.56 (95% CI: 1.15–2.11), respectively (Table 4).

### Exploratory subgroup analysis and evaluation of interaction effects between VMBG and FMBG

Stratified and interactive exploratory analyses were performed across demographic and clinical subgroups, including age, sex, race, severity scores, and relevant comorbidities, to evaluate the robustness of associations between VMBG, FMBG and 30-day mortality in trauma patients (Fig 7). All analyses were conducted within fully adjusted model 3. The stratified analyses consistently demonstrated similar associations of VMBG and FMBG with mortality across most subgroups. Notably, the impact of VMBG on prognosis was more pronounced in patients aged <65 years (P for interaction = 0.008), non-White individuals (P for interaction = 0.001), and those without diabetes (P for interaction<0.001). Similar interaction patterns were observed for FMBG in these subgroups (P for interaction<0.001, 0.025, and 0.002, respectively). However, no significant interactions were detected for either VMBG or FMBG with sex, severity scores, or other comorbidities (all P for interaction>0.05).

## Discussion

This retrospective analysis of clinical data from the MIMIC-IV database demonstrated significant associations between different time intervals of VMBG, FMBG, and 30-day mortality in trauma patients. Under acute critical conditions, activation of the central nervous system (CNS) and neuroendocrine axis triggered the release of effector hormones, including epinephrine, norepinephrine, growth hormone, and glucagon. Excessive catecholamine production enhanced glycogenolysis and hepatic glucose production, stimulated glucagon secretion, and impaired insulin-mediated glucose uptake. Concurrently, these hormones induced the release of proinflammatory cytokines, resulting in systemic inflammatory responses and exacerbating insulin resistance [17,18]. These stress-induced metabolic disturbances ultimately manifest as hyperglycemia and increased glycemic variability [19]. In our study, the mortality group exhibited significantly higher mean glucose levels (p<0.001) and greater glycemic variability indices (p=0.003) compared to the survival group, consistent with previous findings [20].

The debate surrounding the impact of hyperglycemia on critically ill patients had persisted for over a decade. Several studies suggested that hyperglycemia may represent an adaptive response to stress. In septic patients, stress hyperglycemia had been associated with reduced ICU mortality [21], with no observed increase in adverse outcomes even at severe hyperglycemic levels. This protective effect might be attributed to elevated peripheral glucose concentrations facilitating cellular glucose supply [22]. However, Lesur et al. demonstrated differential stress responses between septic and non-septic patients during early ICU admission [23]. In trauma populations, admission hyperglycemia showed significant correlation with initial serum IL-6 levels, serving as a predictor for both injury severity and poor hospitalization outcomes

**Table 4. Cox regression analysis of VMBG, FMBG and 30-day mortality following PSM.**

| VMBG | HR(95%CI) | P | FMBG | HR(95%CI) | P |
|---|---|---|---|---|---|
| <88.1 mg/dl | 2.892(1.162,7.199) | 0.022 | <95.4 mg/dl | 2.380(1.325,4.275) | 0.004 |
| 88.1-125.4 mg/dl | 1(reference) | – | 95.4-134.0 mg/dl | 1(reference) | – |
| ≥125.4 mg/dl | 2.505(1.891,3.318) | <0.001 | ≥134.0 mg/dl | 1.555(1.204,2.008) | <0.001 |
| continuous | 1.012(1.009,1.015) | <0.001 | continuous | 1.005(1.002,1.008) | 0.001 |

PSM: propensity score matching.

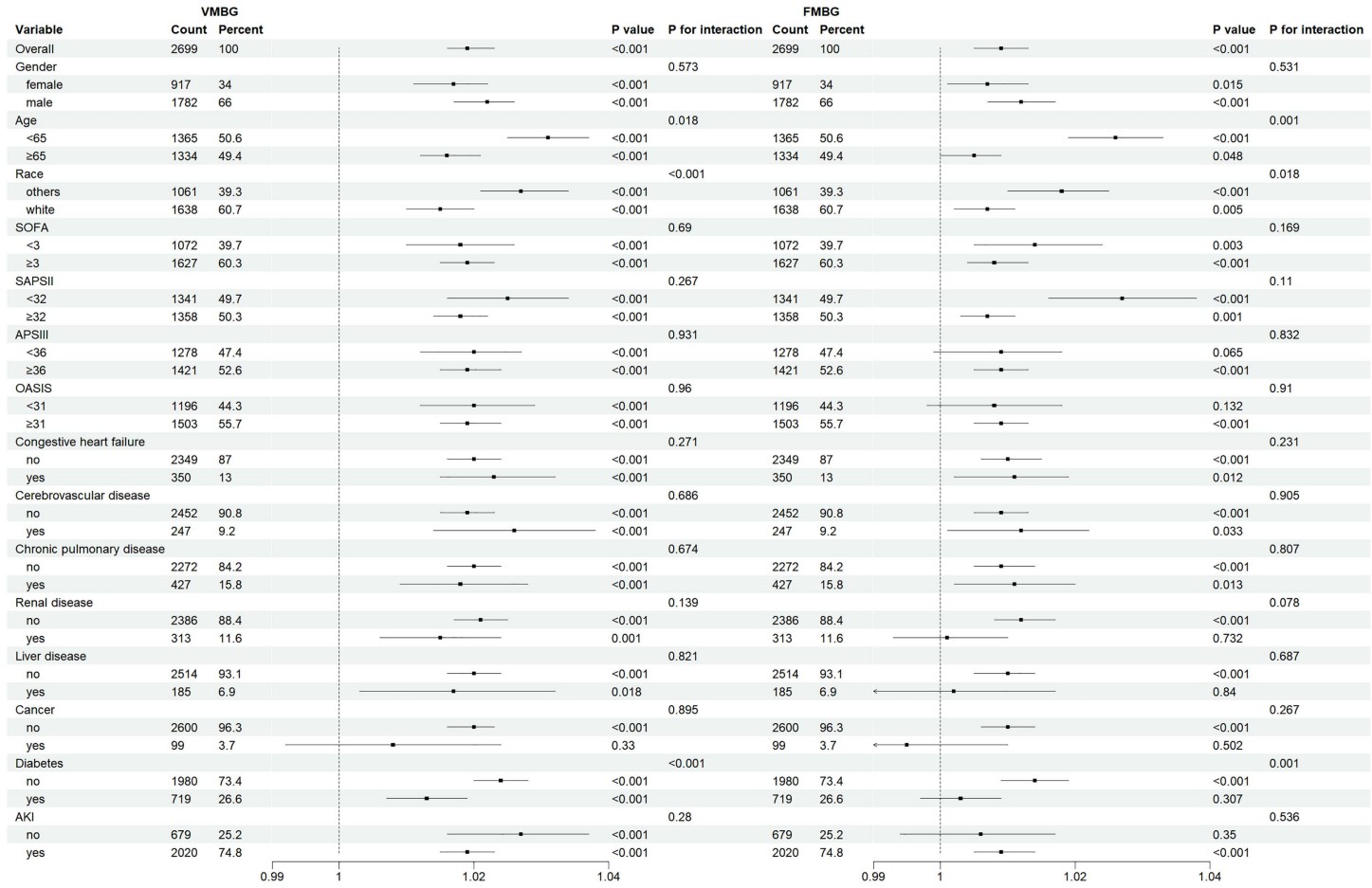

**Fig 7. Forest plots for subgroup analyses of VMBG and FMBG with 30-day mortality.**

[10]. While Rugg et al. recently proposed that stress hyperglycemia might confer benefits in life-threatening acute conditions [6], their study emphasized the need for further investigation into the detrimental effects of sustained hyperglycemia in trauma patients. Importantly, the potential risks of hypoglycemia should not be overlooked in critically ill patients [24]. Kreutziger et al. revealed that both prehospital hypoglycemia and hyperglycemia were associated with traumatic shock, with distinct age-related patterns: hyperglycemia predominated in younger patients versus hypoglycemia in elderly cohorts [25]. Current evidence suggested a non-linear relationship between blood glucose levels and clinical risks in critical care. Furthermore, the majority of studies had employed a standardized blood glucose threshold without considering variations in measurement techniques. However, given the complex and dynamic nature of clinical settings, diverse monitoring methods had become indispensable. Capillary glucose testing predominated in pre-hospital and emergency department settings, while venous and arterial sampling were more common in hospital wards and intensive care units (ICUs). Research had demonstrated significant discrepancies in blood glucose readings obtained through different measurement methods among critically ill patients [14]. Therefore, it might be necessary to adjust the threshold values based on specific clinical contexts.

To further investigate the relationship between glucose levels and clinical outcomes, we performed RCS analyses based on Cox regression models, examining both VMBG and FMBG measurements. The results revealed a significant

J-shaped nonlinear association between VMBG, FMBG, and 30-day mortality. Following the methodology of Wang et al. [26], the intersection point at HR = 1 was identified as the threshold, thereby enabling the categorization of patients into hypoglycemic, normoglycemic, and hyperglycemic groups based on VMBG and FMBG threshold, respectively. The threshold points for VMBG were 88.1 mg/dL and 125.4 mg/dL, while for FMBG they were 95.4 mg/dL and 134.0 mg/dL. Previous studies had demonstrated that extreme hematocrit levels significantly affect capillary glucose measurements [14], with elevated capillary glucose concentrations observed in shock patients receiving norepinephrine treatment or experiencing reduced tissue perfusion. Consistent with these findings, our study found that overall FMBG levels were higher than VMBG levels, and the RCS-derived thresholds for FMBG were also elevated compared to VMBG. Using the VMBG thresholds, patients were stratified into three groups with the intermediate group (88.1–125.4 mg/dL) as reference. Patients with VMBG <88.1 mg/dL and ≥125.4 mg/dL exhibited 2.80-fold and 3.13-fold increased mortality risks, respectively. Similarly, when applying FMBG thresholds with the intermediate group (95.4–134.0 mg/dL) as reference, patients with FMBG <95.4 mg/dL and ≥134.0 mg/dL showed approximately 1.91-fold higher mortality risks. PSM was utilized to further control for potential confounders and to ensure that all included factors were balanced between the mortality and survival groups. Subsequent analyses incorporating VMBG and FMBG as both continuous and categorical variables demonstrated statistically significant associations with outcomes, with hazard ratios (HRs) consistent with multivariable regression results. These findings underscore hyperglycemia as a critical risk factor for mortality in trauma patients, while also highlighting the detrimental impact of hypoglycemia on clinical outcomes. Nevertheless, the identified glucose thresholds, while illuminating a clear nonlinear "J-shaped" relationship, were data-driven, exploratory, and require external validation.

A large-scale observational study [27] demonstrated that patients with mild hypoglycemia (72–81 mg/dL) had significantly higher mortality rates compared to those with normal blood glucose levels, and the severity of hypoglycemia was positively correlated with mortality risk, independent of disease severity. In critically ill patients, hypoglycemia could occur spontaneously or as a consequence of intensive insulin therapy [28]. Given that glucose served as the primary energy source for the brain and cerebral glucose storage capacity was limited, hypoglycemia exerted particularly significant effects on the central nervous system. It could lead to early cerebral hyperemia, progressing to punctate necrosis or cerebral edema, and ultimately resulting in irreversible neuronal damage or even death, significantly impacting patient prognosis [29]. Our study found that, whether classified by VMBG or FMBG criteria, the hypoglycemic group exhibited significantly higher mortality risk than the intermediate glucose group, though slightly lower than the hyperglycemic group. Although literature suggested that capillary blood glucose measurements were susceptible to various interfering factors [30], our data confirmed that it retained certain clinical predictive value. Notably, the normal reference ranges corresponding to different blood glucose monitoring methods were not fixed. In resource-limited settings, when continuous venous or arterial blood glucose monitoring was unavailable, continuous fingertip glucose surveillance could be employed, but with appropriately adjusted glycemic targets to reduce the incidence of hypoglycemia and enhance therapeutic safety.

We analyzed the average glucose levels within various periods following ICU admission to investigate the impact of continuous glucose monitoring on prognosis, in order to reduce reverse causality, temporal overlap, and immortal time bias. We observed that within 24 hours, the VMBG presented only a single risk threshold, whereas the FMBG had already demonstrated typical dual risk thresholds of low and high. This discrepancy was likely attributable to their distinct physiological responses in the acute pathological setting. Venous blood glucose primarily reflected the systemic hyperglycemic load driven by a severe stress response such as a surge in catecholamine and cortisol [10]. In contrast, fingertip blood glucose was more susceptible to local interference under conditions of early microcirculatory instability and potentially compromised peripheral perfusion [14]. Consequently, the earlier emergence of a low-glucose threshold in FMBG could have served as an early warning signal of peripheral hypoperfusion. Through a more rigorous landmark analysis, it was discovered that even after excluding patients who were lost to follow-up within the early 2–3 days (including those who passed away or were discharged), the hyperglycemia at the onset of hospitalization (regardless of whether it was

measured by venous blood or fingertip blood) remained an independent risk factor for 30-day mortality. This indicated that early hyperglycemia was not simply a "consequence" of severe injuries, but rather a modifiable factor that might continuously contribute to the deterioration of the condition. Owing to the limitations of the database records and the intrinsic characteristics of retrospective analysis, we were unable to control or standardize the specific clinical circumstances during blood glucose measurement. For example, in situations where patients were in shock, receive high–dose vasoactive drug support, or suffer from severe hypothermia, fingertip blood glucose may not precisely represent the core blood glucose level of the body, thus introducing measurement bias. This could be one of the potential mechanisms contributing to the disparities in prognostic efficacy and thresholds between fingertip blood glucose and venous blood glucose in this study. When interpreting fingertip blood glucose values, particularly when utilizing them for clinical decision–making, the patient's peripheral tissue microcirculation perfusion at that time should still be taken into account. With time, the risk models revealed by the two measurement methods converged, both demonstrating a clear 'J-shaped' dual-threshold association. As the observation time window was extended, the predictive efficacy improved, which was likely to reflect the increased prognostic information contained in "cumulative glycemic load". These findings were predictive association rather than causal, and prospective intervention studies were needed to verify the causal relationship.

Whether in the early-stage or the medium to long-term, the statistically significant moderate AUC values for both the VMBG and the FMBG suggested that neither can serve as a perfect standalone predictor. Our research results retained a certain degree of clinical significance. It established VMBG as the superior prognostic tool to guide monitoring choices; revealed that early FMBG thresholds may offer unique, complementary warning signals; and identified VMBG as the optimal variable for future integrated risk models where high precision is required. Thus, our work advances prognosis by refining the components and interpretation of metabolic assessment, not by providing a single diagnostic test.

In exploratory subgroup analyses, we observed potential effect between VMBG and FMBG with age, race, and diabetes comorbidity: the impact of elevated glucose levels on prognosis was more pronounced in subgroups of patients aged <65 years, non-White individuals, and those without diabetes. These findings aligned with previous studies [31] indicating that hyperglycemia was more strongly associated with all-cause mortality in younger patients, potentially due to their higher susceptibility to insulin resistance [20,32]. Further analyses suggested distinct pathophysiological mechanisms underlying hyperglycemia in non-diabetic and diabetic patients: non-diabetic hyperglycemia was often related to stress responses, reflecting progressive physiological dysregulation, and its detrimental effects might exceed those of chronic hyperglycemia in diabetic patients [10,33]. Additionally, diabetic patients exhibited greater tolerance to hyperglycemia compared to non-diabetic individuals [34]. An analysis of 95,764 trauma patients revealed that hyperglycemia was associated with increased mortality risk in both diabetic and non-diabetic populations, with the highest mortality observed in non-diabetic hyperglycemic patients [11]. Although current evidence had not clearly delineated differences in hyperglycemia-related risks between White and non-White populations, such disparities might be attributed to variations in metabolic responses, disease susceptibility, healthcare accessibility, and socioeconomic factors among racial groups [35–38], necessitating further investigation. In summary, age, diabetes status, and race might significantly modify the prognostic value of glucose measurements in trauma patients. Venous blood glucose was less influenced by local factors and could more precisely reflect the actual blood glucose level within the body. The subgroup analysis demonstrated that irrespective of whether the patient developed AKI, the venous blood glucose level was significantly associated with the risk of death, indicating a severe stress response and metabolic disorders. While FMBG was associated with the 30-day mortality rate in the AKI group, there was no apparent correlation in the non-AKI group. AKI was not merely a renal function impairment but also signified a systemic severe inflammatory response, endothelial damage, and microcirculation disorder [39]. At this juncture, the value of fingertip blood glucose not only represented the blood glucose concentration but also encompassed information regarding poor tissue perfusion and cellular metabolic disorder [30], thereby endowing it with a stronger prognostic signal. This instability also indirectly validated that the value of fingertip blood glucose as a predictor was not as significant as that of venous blood glucose. Given that this was an exploratory subgroup analysis,

these results were hypothesis–generating and had not been corrected for multiple comparisons. Therefore, they needed to be independently verified in future studies.

This study had several limitations. First, constrained by the database structure, the Injury Severity Score (ISS) was not included, precluding subgroup stratification based on anatomical injury sites and severity levels. Second, limited availability of arterial blood glucose data prevented its incorporation into the multidimensional glucose assessment framework. Third, the lack of standardized glucose measurement frequency of our study, as sicker patients indeed underwent more frequent testing, which could introduce surveillance bias and potentially overestimate the association between glucose indices and mortality. Future prospective studies employing standardized measurement protocols were required to validate our findings. Fourth, while known confounders were adjusted through multivariate models, unmeasured variables might still influence the outcomes. Finally, since this is a retrospective observational study, the causal association between blood glucose and prognosis requires further verification via prospective intervention trials.

In conclusion, both venous and fingertip mean blood glucose levels across various time intervals exhibited nonlinear associations with 30-day mortality among ICU trauma patients, with both hypoglycemia and hyperglycemia significantly increasing mortality risk. Although venous blood glucose monitoring typically demonstrated higher prognostic predictive accuracy compared to fingertip measurements, the early–warning value of fingertip blood glucose should not be overlooked. In clinical monitoring, the characteristics of both measurement methods should be comprehensively recognized. Furthermore, particular attention should be paid to hyperglycemia management in younger patients (<65 years), non-diabetic individuals, and non-White trauma populations to optimize their clinical outcomes.

## Supporting information

**S1 Table. Comparison of VMBG and FMBG between mortality group and survival group at various time intervals.**
VMBG: mean blood glucose of venous. FMBG: mean blood glucose of fingertip.
(DOCX)

**S2 Table. Multivariate COX regression analysis of VMBG and FMBG at different time intervals.** VMBG: mean blood glucose of venous. FMBG: mean blood glucose of fingertip. Adjusted for sex, age, race, comorbidity index, cerebrovascular disease, liver disease, chronic pulmonary disease, diabetes, congestive heart failure, cancer, renal disease, CRRT, ventilation, insulin, transfusion, SOFA, GCS, AKI stage, SAPSII, APSIII, OASIS.
(DOCX)

**S3 Table. The comparison between VMBG, FMBG and 30-day mortality of the landmark cohorts.**
(DOCX)

**S4 Table. Multivariate Cox regression analysis of VMBG, FMBG and 30-day mortality of the Landmark cohorts.** Adjusted for sex, age, race, comorbidity index, cerebrovascular disease, liver disease, chronic pulmonary disease, diabetes, congestive heart failure, cancer, renal disease, CRRT, ventilation, insulin, transfusion, SOFA, GCS, AKI stage, SAPSII, APSIII, OASIS.
(DOCX)

**S5 Table. Baseline characteristics of the complete dataset.** VMBG: mean blood glucose of venous within 30 days. FMBG: mean blood glucose of fingertip within 30 days. VGV: glycemic variability of venous. FGV: glycemic variability of fingertip. SBP: systolic blood pressure. DBP: diastolic blood pressure. MAP: mean arterial pressure. WBC: white blood cell. RBC: red blood cell. RDW: red cell distribution width. INR: international normalized ratio. PT: prothrombin time. PTT: partial thromboplastin time. GCS: Glasgow Coma Scale. SOFA: Sequential Organ Failure Assessment. SAPS II: Simplified Acute Physiology Scores II. APS III: Acute Physiology Score III. OASIS: Oxford Acute Severity of Illness Score. AKI stage: acute kidney injury stage. CRRT: continuous renal replacement therapy.
(DOCX)

**S6 Table. COX regression analysis of VMBG and FMBG in the complete dataset.** VMBG: mean blood glucose of venous. FMBG: mean blood glucose of fingertip. Model1: adjusted for sex, age, race. Model2: adjusted for sex, age, race, comorbidity index, cerebrovascular disease, liver disease, chronic pulmonary disease, diabetes, congestive heart failure, cancer, renal disease. Model3: adjusted for sex, age, race, comorbidity index, cerebrovascular disease, liver disease, chronic pulmonary disease, diabetes, congestive heart failure, cancer, renal disease, CRRT, ventilation, insulin, transfusion, SOFA, GCS, AKI stage, SAPSII, APSIII, OASIS.
(DOCX)

**S7 Table. The comparison between VMBG and FMBG within 30 days without trimming.**
(DOCX)

**S8 Table. Multivariate Cox regression analysis of VMBG, FMBG and 30-day mortality without trimming.** Adjusted for sex, age, race, comorbidity index, cerebrovascular disease, liver disease, chronic pulmonary disease, diabetes, congestive heart failure, cancer, renal disease, CRRT, ventilation, insulin, transfusion, SOFA, GCS, AKI stage, SAPSII, APSIII, OASIS.
(DOCX)

**S9 Table. Baseline characteristics after PSM.** VMBG: mean blood glucose of venous. FMBG: mean blood glucose of fingertip. SBP: systolic blood pressure. DBP: diastolic blood pressure. MAP: mean arterial pressure. WBC: white blood cell. RBC: red blood cell. RDW: red cell distribution width. INR: international normalized ratio. PT: prothrombin time. PTT: partial thromboplastin time. GCS: Glasgow Coma Scale. SOFA: Sequential Organ Failure Assessment. SAPS II: Simplified Acute Physiology Scores II. APS III: Acute Physiology Score III. OASIS: Oxford Acute Severity of Illness Score. AKI stage: acute kidney injury stage. CRRT: continuous renal replacement therapy. PSM: propensity score matching.
(DOCX)

**S1 Fig. Number of venous glucose measurements within 30 day (A).** Number of fingertip glucose measurements within 30 day (B).
(TIF)

**S2 Fig. RCS analysis of the correlation between VMBG, FMBG of various early-time intervals and 30-day mortality.** VMBG: mean blood glucose of venous. FMBG: mean blood glucose of fingertip. RCS: restricted cubic splines.
(TIF)

**S3 Fig. RCS analysis of the correlation between VMBG, FMBG of various medium- and long-term intervals and 30-day mortality.** VMBG: mean blood glucose of venous. FMBG: mean blood glucose of fingertip. RCS: restricted cubic splines.
(TIF)

**S4 Fig. SMDs of variables prior to and following the matching process with a caliper width of 0.2.** SMDs: standardized mean differences.
(TIFF)

**S1 File. Raw data of fingertip glucose within 30 days.**
(raw data of fingertip glucose within 30 days.CSV)

**S2 File. Raw data of venous glucose within 30 days.**
(raw data of venous glucose within 30 days.CSV)

## Author contributions

**Conceptualization:** Fei Yin, Xiaofei Wu, Yuzhou Xu.

**Data curation:** Fei Yin, Zhenguo Qiao.

**Formal analysis:** Fei Yin, Xiaofei Wu.

**Funding acquisition:** Fei Yin.

**Investigation:** Fei Yin, Zhenguo Qiao.

**Methodology:** Fei Yin.

**Project administration:** Fei Yin, Yun Liu.

**Resources:** Fei Yin, Yuzhou Xu.

**Software:** Fei Yin, Yuzhou Xu.

**Supervision:** Fei Yin, Yun Liu.

**Validation:** Fei Yin.

**Visualization:** Fei Yin.

**Writing – original draft:** Fei Yin.

**Writing – review & editing:** Fei Yin, Yun Liu.

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
