## [Decision Letter · Decision Letter 0]

29 Oct 2025

Dear Dr. Liu,

Thank you for submitting your manuscript to PLOS ONE. After careful consideration, we feel that it has merit but does not fully meet PLOS ONE’s publication criteria as it currently stands. Therefore, we invite you to submit a revised version of the manuscript that addresses the points raised during the review process.

We look forward to receiving your revised manuscript.

Kind regards,

Aleksandra Klisic

Academic Editor

PLOS ONE

[The project was supported by the 2024 Excellence Program for Young and Middle-aged Talents at Suzhou Ninth People's Hospital.].

Please respond by return e-mail so that we can amend your financial disclosure and competing interests on your behalf.

Additional Editor Comments (if provided):

Reviewers' comments:

Reviewer's Responses to Questions

**Comments to the Author**

1. Is the manuscript technically sound, and do the data support the conclusions?

Reviewer #1: Yes

Reviewer #2: Yes

2. Has the statistical analysis been performed appropriately and rigorously?

Reviewer #1: I Don't Know

Reviewer #2: Yes

3. Have the authors made all data underlying the findings in their manuscript fully available?

Reviewer #1: Yes

Reviewer #2: Yes

4. Is the manuscript presented in an intelligible fashion and written in standard English?

Reviewer #1: Yes

Reviewer #2: Yes

Reviewer #1: The chosen topic is highly relevant in both the ICU and trauma settings, as the research team clearly demonstrated in the introduction. The work is well described, with clear definitions of some concepts.

Reviewer #2: Dear authors, in the conclusion of your study you emphasized the limits of this study and I fully agree with what was stated. Please clarify the glycemic profiling and the comparative time of sampling, I cannot establish from your data whether these are average values or the highest values, nor do I have insight into the timeline of sampling.

**Do you want your identity to be public for this peer review?** For information about this choice, including consent withdrawal, please see our Privacy Policy

Reviewer #1: **Yes:** Felipe Martins Liporaci

Reviewer #2: **Yes:** Snezana Rsovac

---

## [Author Response · Author response to Decision Letter 1]

4 Nov 2025

Dear editor and reviewers of PLOS ONE:

Our reference: PONE-D-25-23041

Title: Differential analysis of mean blood glucose levels from venous and fingertip in predicting 30-day mortality among ICU patients with severe trauma: A retrospective study utilizing the MIMIC-IV database

By: Yun Liu et al

Thank you very much for your letter and for the editors’ and reviewers’ comments concerning our manuscript entitled "Differential analysis of mean blood glucose levels from venous and fingertip in predicting 30-day mortality among ICU patients with severe trauma: A retrospective study utilizing the MIMIC-IV database" (ID: PONE-D-25-23041). These comments are of great reference value to the revision and improvement of our paper and have important guiding significance to our researches. We have studied comments carefully and have made correction. We hope that the revision is acceptable and look forward to hearing from you soon. Revised portion are marked in color in the paper. The main corrections in the paper and the responds to the journal requirements and reviewer’s comments are as flowing:

We confirm that the manuscript complies with the style requirements of PLOS ONE, including the specified file naming conventions.

[The project was supported by the 2024 Excellence Program for Young and Middle-aged Talents at Suzhou Ninth People's Hospital.].

Please respond by return e-mail so that we can amend your financial disclosure and competing interests on your behalf.

Our research was supported by the 2024 Excellence Program for Young and Middle-aged Talents at Suzhou Ninth People's Hospital. We confirm that the funders had no role in study design, data collection and analysis, decision to publish, or preparation of the manuscript.

3.PLOS requires an ORCID iD for the corresponding author in Editorial Manager on papers submitted after December 6th, 2016. Please ensure that you have an ORCID iD and that it is validated in Editorial Manager. To do this, go to ‘Update my Information’ (in the upper left-hand corner of the main menu), and click on the Fetch/Validate link next to the ORCID field. This will take you to the ORCID site and allow you to create a new iD or authenticate a pre-existing iD in Editorial Manager.

The corresponding author had registered an ORCID iD and successfully updated it in the Editorial Manager of PLOS. The ORCID iD is: https://orcid.org/0009-0005-8521-8723.

The reviewer comments did not indicate the need to cite of specific content from previously published works.

We have conducted a thorough review of the reference list. We can confirm that all references are complete and correct, and to the best of our knowledge, no retracted literature has been cited.

Reviewers' comments:

Reviewer #1: The chosen topic is highly relevant in both the ICU and trauma settings, as the research team clearly demonstrated in the introduction. The work is well described, with clear definitions of some concepts.

We would like to express our sincere gratitude to Reviewer #1 for their generous and positive assessment of our work. We are greatly encouraged by their comments that the chosen topic is highly relevant and that the work is well-described with clear definitions. Such acknowledgement is very motivating for our team.

Reviewer #2: Dear authors, in the conclusion of your study you emphasized the limits of this study and I fully agree with what was stated. Please clarify the glycemic profiling and the comparative time of sampling, I cannot establish from your data whether these are average values or the highest values, nor do I have insight into the timeline of sampling.

During the revision process, we incorporated the original data files for venous and fingertip glucose measurements, specifically "raw data of venous glucose within 30 days.csv" and "raw data of fingertip glucose within 30 days.csv." These datasets included information on 2,699 trauma patients, including patient IDs, ICU admission times, all recorded venous glucose values within 30 days, fingertip glucose values within 30 days, and corresponding glucose measurement timestamps. The data were organized sequentially by patient ID and measurement timestamp, enabling a clear and systematic review of each glucose value and its associated time point. As illustrated in Figures R1 and R2, fingertip glucose measurements exhibited a more dispersed distribution across daily intervals, whereas venous glucose measurements were predominantly clustered between 0:00 and 7:00. Further analysis revealed that there was no significant difference in the number of venous glucose measurements between non-survivors and survivors groups within 30 days; however, the number of fingertip glucose measurements in the non-survivor group was higher compared to the survivor group, and both venous and fingertip glucose monitoring frequencies were elevated in the non-survivor group (Table R1). This pattern indicated that glucose monitoring frequency might be determined by clinical necessity. After adjusting for glucose measurement frequency using Model 3, the associations between VMBG and FMBG with 30-day mortality remained statistically significant. Specifically, each 1-unit increase in VMBG was associated with a 1.019-fold increase in mortality risk (HR = 1.019, 95% CI: 1.015–1.022), and each 1-unit increase in FMBG corresponded to a 1.009-fold increase in risk (HR = 1.009, 95% CI: 1.004–1.012) (Tables R2 and R3).

Due to the inherent limitations of retrospective studies, blood glucose measurements were characterized by non-standardized timing and variable resolution. To mitigate potential bias arising from inconsistent measurement protocols, we conducted additional analyses to evaluate the predictive performance of mean blood glucose levels across different time intervals—specifically, within 24 hours, 5 days, 10 days, and 20 days—on 30-day mortality. These analyses included both intergroup and intragroup comparisons. The results were presented in the section titled "The associations between VMBG, FMBG, and 30-day mortality across various time intervals," along with Fig. 4 and Supplementary Table 4.

Furthermore, the VMBG reported in the manuscript represents the mean of all venous blood glucose measurements for each patient over a 30-day period, while the FMBG refer to the mean of all fingertip blood glucose measurements collected during the same period. This clarification had been incorporated into the Patients section under "Methods" to enhance transparency and consistency.

---

## [Decision Letter · Decision Letter 1]

12 Jan 2026

Dear Dr. Liu,

Thank you for submitting your manuscript to PLOS ONE. After careful consideration, we feel that it has merit but does not fully meet PLOS ONE’s publication criteria as it currently stands. Therefore, we invite you to submit a revised version of the manuscript that addresses the points raised during the review process.

https://journals.plos.org/plosone/s/submission-guidelines#loc-laboratory-protocols . Additionally, PLOS ONE offers an option for publishing peer-reviewed Lab Protocol articles, which describe protocols hosted on protocols.io. Read more information on sharing protocols at https://plos.org/protocols?utm_medium=editorial-email&utm_source=authorletters&utm_campaign=protocols .

We look forward to receiving your revised manuscript.

Kind regards,

Aleksandra Klisic

Academic Editor

PLOS One

**Journal Requirements:**

Reviewers' comments:

Reviewer's Responses to Questions

**Comments to the Author**

Reviewer #1: All comments have been addressed

Reviewer #3: All comments have been addressed

2. Is the manuscript technically sound, and do the data support the conclusions?

Reviewer #1: Yes

Reviewer #3: Yes

3. Has the statistical analysis been performed appropriately and rigorously?

Reviewer #1: I Don't Know

Reviewer #3: Yes

4. Have the authors made all data underlying the findings in their manuscript fully available?

Reviewer #1: Yes

Reviewer #3: Yes

5. Is the manuscript presented in an intelligible fashion and written in standard English?

Reviewer #1: Yes

Reviewer #3: Yes

**Reviewer #1:**  (No Response)

**Reviewer #3:** My Report:

I commend the authors for addressing an important and clinically meaningful question regarding differential prognostic value of venous versus fingertip mean blood glucose in critically ill trauma patients. The large sample size, comprehensive statistical framework, and multiple sensitivity analyses are notable strengths. Nevertheless, several methodological and reporting issues in the Materials and Methods and Results sections should be addressed to improve transparency, reproducibility, and interpretability.

1. The definition of VMBG and FMBG as mean glucose values over the same 30-day period used for the primary outcome introduces potential temporal overlap between exposure and outcome. This raises concerns regarding reverse causation and immortal time bias. Additional clarification or justification is needed, and stronger emphasis on early-interval analyses is recommended.

2. The lack of standardized glucose measurement frequency may bias mean glucose estimates, particularly since sicker patients are more likely to undergo frequent testing. This issue should be more explicitly acknowledged and discussed as a methodological limitation.

3. Although glycemic variability is defined and reported, its role in the analytical framework is unclear, as it is not incorporated into the primary multivariable models. A clearer rationale for its inclusion or exclusion is warranted.

4. The choice of a relatively wide caliper (0.4) requires justification. While preserving sample size is important, readers would benefit from an explanation of the balance–precision trade-off underlying this decision.

5. Although statistically significant differences between VMBG and FMBG are observed, both AUCs indicate only moderate discrimination. A brief comment on clinical relevance versus statistical significance would strengthen the results interpretation.

6. The identification of J-shaped associations using RCS is a major strength. However, the derived glucose thresholds are data-driven and should be reported with caution, emphasizing their exploratory rather than prescriptive nature.

7. The observed interactions with age, race, and diabetes status are interesting but should be interpreted cautiously due to multiple comparisons. These findings should be clearly labeled as exploratory.

8. The increasing predictive performance with longer glucose observation windows is expected but further highlights temporal overlap concerns. These results are best framed as predictive rather than causal.

**Do you want your identity to be public for this peer review?** For information about this choice, including consent withdrawal, please see our Privacy Policy

Reviewer #1: **Yes:** Felipe Martins Liporaci

Reviewer #3: **Yes:** Fateme Binayi

---

## [Author Response · Author response to Decision Letter 2]

24 Jan 2026

Dear editor and reviewers of PLOS ONE:

Our reference: PONE-D-25-23041R1

Title: Differential analysis of mean blood glucose levels from venous and fingertip in predicting 30-day mortality among ICU patients with severe trauma: A retrospective study utilizing the MIMIC-IV database

By: Yun Liu et al

Thank you very much for your letter and for the editors’ and reviewers’ comments concerning our manuscript entitled "Differential analysis of mean blood glucose levels from venous and fingertip in predicting 30-day mortality among ICU patients with severe trauma: A retrospective study utilizing the MIMIC-IV database" (ID: PONE-D-25-23041R1). These comments are of great reference value to the revision and improvement of our paper and have important guiding significance to our researches. We have studied comments carefully and have made correction. We hope that the revision is acceptable and look forward to hearing from you soon. Revised portion are marked in color in the paper. The main corrections in the paper and the responds to the journal requirements and reviewer’s comments are as flowing:

Reviewers' comments:

Reviewer #3:

1. The definition of VMBG and FMBG as mean glucose values over the same 30-day period used for the primary outcome introduces potential temporal overlap between exposure and outcome. This raises concerns regarding reverse causation and immortal time bias. Additional clarification or justification is needed, and stronger emphasis on early-interval analyses is recommended.

We sincerely thank the reviewer for raising the crucial methodological point regarding temporal overlap, reverse causation and immortal time bias. In the revised version, we had presented a comprehensive description of the correlation analysis between early-interval glucose exposure and prognosis. The main research had been modified to: To compare the predictive efficacy of mean blood glucose of venous (VMBG) and mean blood glucose of fingertip (FMBG) in assessing the 30-days mortality risk of ICU trauma patients, and to systematically evaluate the prognostic value of early - stage (24-hour, 2-day, 3-day) and long - term (30-day) monitoring intervals. Specifically, we had performed new multivariable analyses demonstrating that the mean blood glucose levels during the initial 24-hour, 2-day, and 3-day periods following admission were also significantly associated with 30-day mortality (Table 1).

We had highlighted the research findings from the early-interval glucose monitoring windows (24-hour, 2-day, and 3-day) because these results provide greater methodological advantages in avoiding time overlap. We presented the information in the section titled "The associations between VMBG, FMBG of various time intervals and 30-day mortality". We also observed that within 24 hours, the VMBG presented only a single risk threshold, whereas the FMBG had already demonstrated typical dual risk thresholds of low and high (Fig 1). This discrepancy was likely attributable to their distinct physiological responses in the acute pathological setting. VMBG primarily reflected the systemic hyperglycemic load driven by a severe stress response (e.g., catecholamine and cortisol surge) [1]. In contrast, FMBG was more susceptible to local interference under conditions of early microcirculatory instability and potentially compromised peripheral perfusion. It had been documented that, compared to arterial levels, FMBG may significantly underestimate blood glucose in patients with shock or on vasopressors [2]. Consequently, the earlier emergence of a low-glucose threshold in FMBG could have served as an early warning signal of peripheral hypoperfusion. With time, the risk models revealed by the two measurement methods converged, both demonstrating a clear ‘J-shaped’ dual-threshold association.

Concurrently, we retained the sensitivity analysis, PSM analysis, and subgroup analysis for VMBG and FMBG of 30-day. Despite a certain degree of temporal overlap, these exploratory analyses, to varying extents, disclosed the correlation between blood glucose and prognosis and maintained a high level of consistency and robustness, thereby offering hypotheses for further research.

2. The lack of standardized glucose measurement frequency may bias mean glucose estimates, particularly since sicker patients are more likely to undergo frequent testing. This issue should be more explicitly acknowledged and discussed as a methodological limitation.

We thank the reviewer for this crucial methodological insight. We fully agree that the lack of standardized glucose measurement frequency is an important limitation of our retrospective study, as sicker patients indeed undergo more frequent testing, which could introduce surveillance bias and potentially overestimate the association between glucose indices and mortality. We have now explicitly acknowledged and discussed this limitation in the revised Discussion/Limitations section. We further note that future prospective studies with standardized measurement protocols are needed to confirm our findings.

3. Although glycemic variability is defined and reported, its role in the analytical framework is unclear, as it is not incorporated into the primary multivariable models. A clearer rationale for its inclusion or exclusion is warranted.

We sincerely thank the reviewer for raising this important point. Our decision not to incorporate glycemic variability (GV) into the primary multivariate models was based on a comprehensive assessment including clinical relevance, statistical metrics, and methodological considerations, as detailed below:

One of the objectives of our study was to directly compare the prognostic value of mean blood glucose from two different sampling sites (venous vs. fingertip). In the exploratory analysis, GV was incorporated into the multivariate model. Although GV also demonstrated statistical significance, it did not significantly alter the HRs and statistical significance of the mean blood glucose variables(Table 2, Table3). Furthermore, as a result of including the variable GV associated with mean blood glucose, the complexity of the model also rose.

We quantified the additional predictive value of GV. As shown in Tables 2 and 3, when comparing Model a2 with Model a3, and Model b2 with Model b3, there were only minimal changes in their Akaike Information Criterion (AIC) and C - index. Additionally, we utilized the survIDINRI package in R to calculate the net reclassification index (NRI) and the integrated discrimination improvement index (IDI) for model comparison. The results indicated that there was no significant improvement in the NRI and IDI of Model a3 relative to Model a2, with values of 0.052 (-0.101 - 0.128) p = 0.252 and 0.003 (-0.001 - 0.010) p = 0.153, respectively. Similarly, there was no significant improvement in the NRI and IDI of Model b3 relative to Model b2, with values of 0.28 (-0.059 - 0.085) p = 0.292 and 0.003 (-0.001 - 0.013) p = 0.106, respectively. Collectively, these indicators suggested that, within the multivariate model of this cohort, neither VGV nor FGV provided substantial additional prognostic information for our primary outcome indicators.

GV was defined as the ratio of the standard deviation (SD) of blood glucose levels to the mean blood glucose (MBG) concentration, and it served as a proportional covariate. When both GV and MBG were incorporated into the regression model simultaneously, although the VIF values wew both less than 10, a certain mathematical coupling exists. For instance, when interpreting the coefficient of MBG, GV was held at a fixed value, and it was necessary to assume that SD varies strictly in a fixed proportion with MBG. This situation did not align with biological reality. As Valle et al. had expressed concerns, the inclusion of proportional covariates in the regression model complicated model interpretation[3]. Furthermore, the fixed proportion of GV could manifest as either high SD/high MBG or low SD/low MBG. It remained to further study whether these two blood glucose patterns exert different physiological and pathological effects.

4. The choice of a relatively wide caliper (0.4) requires justification. While preserving sample size is important, readers would benefit from an explanation of the balance–precision trade-off underlying this decision.

We utilized the nearest neighbor algorithm for propensity score matching (PSM). Although the commonly recommended caliper width was 0.2[4], to optimize the balance–precision trade-off and to prevent a substantial number of sample losses in the limited overlapping areas within our cohort, we selected a matching width of 0.4. The chosen caliper width of 0.4 achieved an excellent balance of covariates while retaining a sufficient quantity of matches, and all the post - matching standardized mean differences (SMDs) were less than 0.1. This balance was verified through a sensitivity analysis employing a 0.2 caliper width (Fig 2).

In the "Statistical analysis" section of the original manuscript, we have modified it to: To optimize the balance–precision trade-off, propensity score matching (PSM) analysis was conducted using a nearest-neighbor matching algorithm with a caliper width of 0.4 at a 2:1 ratio. And a sensitivity analysis was conducted using caliper width of 0.2.

5. Although statistically significant differences between VMBG and FMBG are observed, both AUCs indicate only moderate discrimination. A brief comment on clinical relevance versus statistical significance would strengthen the results interpretation.

We acknowledged that while the statistically significant difference in AUC between VMBG and FMBG was observed, the absolute values indicated only moderate discriminatory power, meaning neither metric was suitable as a standalone, perfect predictor for individual prognosis. Thus, the significance of our work lay in refining monitoring strategies and building a more nuanced foundation for risk assessment, rather than offering a single diagnostic tool.

6. The identification of J-shaped associations using RCS is a major strength. However, the derived glucose thresholds are data-driven and should be reported with caution, emphasizing their exploratory rather than prescriptive nature.

We thank the reviewer for this important point. We agreed the derived glucose thresholds were data-driven. In our revision, we explicitly reported them with caution. We labeled them as exploratory and hypothesis generating, not prescriptive clinical targets. We emphasized the biological insight from the J-shaped pattern over the specific threshold values. These thresholds require external validation but help advance the pathophysiological understanding of dysglycemia in trauma.

7. The observed interactions with age, race, and diabetes status are interesting but should be interpreted cautiously due to multiple comparisons. These findings should be clearly labeled as exploratory.

We fully agreed that the observed interactions with age, race, and diabetes status should be interpreted with great caution. In the revised manuscript, we had clearly designated these findings as exploratory in both the results section and the discussion section. In the discussion section, we have re - elaborated on the interpretation method, indicating that within these subgroups, the relationship between blood sugar and prognosis may be particularly significant and merits specialized future research, rather than being considered as the final conclusion.

8. The increasing predictive performance with longer glucose observation windows is expected but further highlights temporal overlap concerns. These results are best framed as predictive rather than causal.

We appreciated the reviewer's important point. We agreed that as the observation windows became longer and the glucose information accumulated, the predictive performance increased. Although this was expected, it highlighted the core issue of temporal overlap. In response, we reframed these specific findings in the revised Discussion as demonstrating a predictive association rather than implying causality. We contrasted this with our analysis of early glucose windows, which was designed to better support causal hypotheses. Finally, we stated that any causal inference would require prospective interventional trials.

Table 1 Multivariate COX regression analysis of VMBG and FMBG at different time intervals

Time Intervals VMBG FMBG

HR(95%CI) P HR(95%CI) P

within 24 hours (mg/dL) 1.003(1.001,1.006) 0.006 1.004(1.002,1.007) 0.002

within 48 hours (mg/dL) 1.007(1.004,1.010) �0.001 1.007(1.003,1.010) �0.001

within 72 hours (mg/dL) 1.010(1.007,1.013) �0.001 1.009(1.005,1.012) �0.001

within 5 days (mg/dL) 1.013(1.009,1.016) �0.001 1.009(1.005,1.013) �0.001

within 10 days (mg/dL) 1.016(1.012,1.020) �0.001 1.009(1.006,1.013) �0.001

within 20 days (mg/dL) 1.019(1.015,1.022) �0.001 1.009(1.005,1.013) �0.001

within 30 days (mg/dL) 1.019(1.016,1.023) �0.001 1.009(1.006,1.013) �0.001

VMBG: mean blood glucose of venous. FMBG: mean blood glucose of fingertip. Adjusted for sex, age, race, comorbidity index, cerebrovascular disease, liver disease, chronic pulmonary disease, diabetes, congestive heart failure, cancer, renal disease, CRRT, ventilation, insulin, transfusion, SOFA, GCS, AKI stage, SAPSⅡ, APSⅢ, OASIS

Table 2 Exploratory analysis of VMBG and VGV in the multivariate model

Model VMBG VGV AIC C-index

HR(95% CI) P VIF HR(95% CI) P VIF

Model a1 1.018(1.707,2.112) <0.001 1.273 0.996(0.857,1.048) 0.299 1.273 5161.582 0.696

Model a2 1.019(1.016,1.023) <0.001 1.642 - - - 4780.341 0.833

Model a3 1.022(1.018,1.026) <0.001 2.043 0.988(0.978,0.997) 0.011 1.427 4775.493 0.834

Model a1: VMBG adjusted for VGV. Model a2: VMBG adjusted for Covariates. Model a3: VMBG adjusted for VGV and Covariates. The Covariates included sex, age, race, comorbidity index, cerebrovascular disease, liver disease, chronic pulmonary disease, diabetes, congestive heart failure, cancer, renal disease, CRRT, ventilation, insulin, transfusion, SOFA, GCS, AKI stage, SAPSⅡ, APSⅢ, OASIS. VMBG: mean blood glucose of venous within 30 days. VGV: glycemic variability of venous within 30 days. Akaike Information Criterion (AIC). Variance Inflation Factor (VIF).

Table 3 Exploratory analysis of FMBG and FGV in the multivariate model

Model FMBG FGV AIC C-index

HR(95% CI) P VIF HR(95% CI) P VIF

Model b1 1.009(1.006,1.011) <0.001 1.132 1.026(1.015,1.037) <0.001 1.132 5224.042 0.648

Model b2 1.009(1.006,1.013) <0.001 1.637 - - - 4860.390 0.816

Model b3 1.009(1.002,1.013) <0.001 1.658 1.014(1.002,1.028) 0.026 1.383 4857.521 0.816

Model b1: FMBG adjusted for FGV. Model b2: FMBG adjusted for Covariates. Model b3: FMBG adjusted for FGV and Covariates. The Covariates included sex, age, race, comorbidity index, cerebrovascular disease, liver disease, chronic pulmonary disease, diabetes, congestive heart failure, cancer, renal disease, CRRT, ventilation, insulin, transfusion, SOFA, GCS, AKI stage, SAPSⅡ, APSⅢ, OASIS. FMBG: mean blood glucose of fingertip within 30 days. FGV : glycemic variability of fingertip within 30 days. Akaike Information Criterion (AIC). Variance Inflation Factor (VIF).

---

## [Decision Letter · Decision Letter 2]

29 Jan 2026

Dear Dr. Liu,

Thank you for submitting your manuscript to PLOS ONE. After careful consideration, we feel that it has merit but does not fully meet PLOS ONE’s publication criteria as it currently stands. Therefore, we invite you to submit a revised version of the manuscript that addresses the points raised during the review process.

We look forward to receiving your revised manuscript.

Kind regards,

Aleksandra Klisic

Academic Editor

PLOS One

Journal Requirements:

Reviewer's Responses to Questions

**Comments to the Author**

Reviewer #3: All comments have been addressed

2. Is the manuscript technically sound, and do the data support the conclusions?

Reviewer #3: Yes

3. Has the statistical analysis been performed appropriately and rigorously?

Reviewer #3: Yes

4. Have the authors made all data underlying the findings in their manuscript fully available?

Reviewer #3: Yes

5. Is the manuscript presented in an intelligible fashion and written in standard English?

Reviewer #3: Yes

Reviewer #3: My Report:

This manuscript leverages the large, contemporary, and high-quality MIMIC-IV database and applies a comprehensive suite of modern statistical techniques, including time-to-event modeling, restricted cubic splines, propensity score matching, and multiple sensitivity analyses. The systematic comparison between venous and fingertip mean blood glucose as prognostic indicators in ICU trauma patients is both novel and clinically relevant. Overall, the study is methodologically up-to-date, well-powered, and addresses an important gap in the literature regarding modality-specific glycemic assessment in critically injured patients.

Comments

1. The inclusion criterion requiring at least two glucose measurements within 30 days may introduce substantial heterogeneity in the precision of mean blood glucose estimation. Reporting the distribution of glucose measurement frequency per patient would help readers assess the robustness of MBG estimates.

2. Glycemic variability is defined as SD divided by the mean glucose, which introduces intrinsic mathematical coupling with MBG. Although GV was not included in the primary multivariable models, this dependency should be explicitly acknowledged as a methodological limitation.

3. The manuscript does not clarify the clinical conditions under which venous and fingertip glucose measurements were obtained (e.g., shock states, vasopressor use, hypothermia). These factors may systematically bias fingertip glucose values and warrant explicit discussion in the Methods section.

4. Multiple severity-of-illness scores (SOFA, SAPS II, APS III, and OASIS) are simultaneously included in the adjusted models. Given their conceptual overlap, an assessment of collinearity or a justification for retaining all scores concurrently would improve model transparency.

6. Extreme value handling using 1% tail trimming is mentioned, but the specific variables affected and the impact of this decision on effect estimates are not reported. A brief sensitivity assessment would enhance reproducibility.

7. Although early time-window analyses were conducted to mitigate immortal time bias, the application of formal landmark analysis or time-dependent Cox models could further strengthen causal interpretability.

8. While the difference in AUC between VMBG and FMBG is statistically significant, the absolute values indicate only moderate discrimination. A clearer discussion of clinical relevance versus statistical significance would improve result interpretation.

9. The progressive increase in hazard ratios with longer glucose observation windows is expected and likely reflects cumulative prognostic information. These findings should be clearly distinguished as predictive rather than causal.

10. Subgroup analyses by age, race, and diabetes status are intriguing but involve multiple comparisons. These results should be explicitly labeled as hypothesis-generating.

11. Given the known limitations of fingertip glucose measurements under conditions of impaired peripheral perfusion, stratified analyses among patients receiving vasopressors or with severe AKI could further contextualize the findings.

12. The interpretation of ROC analyses alongside survival models would be strengthened by reporting model calibration metrics, such as calibration plots or Brier scores.

**Do you want your identity to be public for this peer review?** For information about this choice, including consent withdrawal, please see our Privacy Policy

Reviewer #3: **Yes:** Fateme Binayi

---

## [Author Response · Author response to Decision Letter 3]

2 Feb 2026

Dear editor and reviewers of PLOS ONE:

Our reference: PONE-D-25-23041R2

Title: Differential analysis of mean blood glucose levels from venous and fingertip in predicting 30-day mortality among ICU patients with severe trauma: A retrospective study utilizing the MIMIC-IV database

By: Yun Liu et al

Thank you very much for your letter and for the editors’ and reviewers’ comments concerning our manuscript entitled "Differential analysis of mean blood glucose levels from venous and fingertip in predicting 30-day mortality among ICU patients with severe trauma: A retrospective study utilizing the MIMIC-IV database" (ID: PONE-D-25-23041R2). We have studied comments carefully and have made correction. We hope that the revision is acceptable and look forward to hearing from you soon. Revised portion are marked in color in the paper. The main corrections in the paper and the responds to the journal requirements and reviewer’s comments are as flowing:

Comments:

1.The inclusion criterion requiring at least two glucose measurements within 30 days may introduce substantial heterogeneity in the precision of mean blood glucose estimation. Reporting the distribution of glucose measurement frequency per patient would help readers assess the robustness of MBG estimates.

All patients underwent a minimum of two venous blood glucose measurements and two fingertip blood glucose measurements during the 30-day observation period. The quantity of venous blood glucose measurements was 9[5, 17], and 2337 cases (86.6%) had four or more measurements. The quantity of fingertip blood glucose measurements was 7[4, 17], and 2154 patients (79.8%) had four or more measurements. The distribution of glucose measurement frequencies for all patients was presented in Figure 1. We had made revisions in the "Results/Baseline characteristics" section, lines 170-174 of the manuscript.

2.Glycemic variability is defined as SD divided by the mean glucose, which introduces intrinsic mathematical coupling with MBG. Although GV was not included in the primary multivariable models, this dependency should be explicitly acknowledged as a methodological limitation.

Within the "Methods/Statistical Analysis" section, lines 149-151, it was explicitly stated that there exists an intrinsic mathematical correlation between MBG and GV. Owing to this methodological constraint, it was not incorporated into the main multivariate model to prevent the emergence of interpretational challenges.

3.The manuscript does not clarify the clinical conditions under which venous and fingertip glucose measurements were obtained (e.g., shock states, vasopressor use, hypothermia). These factors may systematically bias fingertip glucose values and warrant explicit discussion in the Methods section.

We express our sincere gratitude to the reviewers for highlighting this critical point. We fully concur that the clinical status during measurement, such as shock states, vasopressor use, hypothermia, was one of the factor that systematically impacts the accuracy of fingertip blood glucose measurement, and it was necessary to conduct a clear discussion. To address this, we had made the following augmentations in the revised version: In the "Methods/Patients" section, lines 125-127, we had included a clarification indicating that the blood glucose data in this study was sourced from routine clinical records, and the measurement conditions were non - standardized. We also recognized the potential limitations of fingertip glucose in situations like shock, the utilization of vasoactive drugs and hypothermia. In the "Discussion" section, lines 484-492, we had specifically inserted a paragraph to thoroughly explore the potential bias effect of the peripheral perfusion status on the fingertip glucose value and expound on it as one of the mechanisms that may account for the disparity between fingertip and venous glucose results. Meanwhile, we emphasized the significance of taking the patient's circulatory status into account during clinical interpretation.

4.Multiple severity-of-illness scores (SOFA, SAPS II, APS III, and OASIS) are simultaneously included in the adjusted models. Given their conceptual overlap, an assessment of collinearity or a justification for retaining all scores concurrently would improve model transparency.

To assess collinearity, the variance inflation factors (VIF) for the covariates incorporated in the multivariate model (including SOFA, SAPS II, APS III, and OASIS) were calculated (Table 1). The VIF values for all covariates were below 5, suggesting that no severe multicollinearity was present. This diagnostic outcome had been elaborated on in the "Statistical Methods" section, lines 163-165.

6.Extreme value handling using 1% tail trimming is mentioned, but the specific variables affected and the impact of this decision on effect estimates are not reported. A brief sensitivity assessment would enhance reproducibility.

In the "Statistical Methods" section, it had been clearly stated that for continuous variables, a 1% tail trimming was applied to mitigate the impact of extreme values. In response to this concern, an additional sensitivity analysis was conducted: the main model was run using the original data without 1% tail trimming. The results were largely consistent with those of the main analysis, indicating that the conclusions drawn were not sensitive to the treatment of extreme values (Table2 and Table3). The results of this analysis had been included in the "Results/Sensitivity analysis" section, lines 339-342, and the table was presented as supplementary material.

7.Although early time-window analyses were conducted to mitigate immortal time bias, the application of formal landmark analysis or time-dependent Cox models could further strengthen causal interpretability.

To conduct a more rigorous assessment of the association between early blood glucose levels and long-term mortality risk and effectively mitigate the immortal time bias, this study further performed a Landmark analysis. The Landmark time point was designated as the third day subsequent to the patient's admission. A new analysis cohort was established which encompassed all patients who survived until the third day and had not been discharged (n = 2516, accounting for 93.2% of the original cohort). In this cohort, VMBG and FMBG for each patient were calculated within the initial 3 days following admission to the ICU. The results demonstrated that the VMBG and FMBG of the deceased group were significantly higher than those of the survivors (Table 4). Subsequently, these two indicators were respectively utilized as exposure variables. Multivariate Cox proportional hazards regression model was applied to evaluate their association with the mortality risk from the Landmark time point (the third day after admission) to the end of the 30th day. The analysis revealed that both VMBG and FMBG within 3 days were significantly positively correlated with the 30-day mortality risk: the adjusted hazard ratio (HR) for VMBG was 1.006 (95% CI: 1.003–1.010), and the adjusted HR for FMBG was 1.005 (95% CI: 1.001–1.009) (Table 5). When the sensitivity analysis was conducted at the landmark time point designated as the second day after the patient's admission, the results were also encouraging. Since patients with a hospital stay of less than 24 hours had been excluded, the VMBG and FMBG within 24 hours had already been analyzed as the Landmark analysis.The above modifications were located in the "Results" section, lines 312-330 and "Discussion" section, lines 478-484 of the manuscript.

8.While the difference in AUC between VMBG and FMBG is statistically significant, the absolute values indicate only moderate discrimination. A clearer discussion of clinical relevance versus statistical significance would improve result interpretation.

We thank the reviewer for raising this point regarding the clinical interpretation of the AUC values. We addressed this same comment in detail during the previous revision round. As noted in our prior response and in the revised manuscript, we had expanded the interpretation to clearly distinguish statistical significance from clinical utility. In accordance with the current suggestions, we have adjusted the positions of the paragraphs and made further revisions to the wording, thereby enabling readers to perceive this difference more clearly. ("Discussion" section, lines 498-505)

9.The progressive increase in hazard ratios with longer glucose observation windows is expected and likely reflects cumulative prognostic information. These findings should be clearly distinguished as predictive rather than causal.

We thank the reviewers for their clarification. We had clearly distinguished this in the "Discussion" section, lines 494-497. As the observation time window was extended, the predictive efficacy improved, which was likely to reflect the increased prognostic information contained in "cumulative glycemic load". We explicitly stated that these analytical results should be interpreted as a strong predictive association, and the causal explanation requires prospective intervention studies to verify.

10.Subgroup analyses by age, race, and diabetes status are intriguing but involve multiple comparisons. These results should be explicitly labeled as hypothesis-generating.

In the revised version, we had explicitly marked the subgroup analysis sections associated with age, race, and diabetes status as "exploratory subgroup analysis" ("Results" section, lines 374-386). Additionally, in the "Discussion" section, lines 534-536, we had emphasized that these results were hypothesis - generating and had not undergone correction for multiple comparisons. Consequently, they required independent verification in future studies.

11. Given the known limitations of fingertip glucose measurements under conditions of impaired peripheral perfusion, stratified analyses among patients receiving vasopressors or with severe AKI could further contextualize the findings.

Owing to the limitations of the retrospective study grounded in the database, we chose the AKI indicator to perform a stratified analysis of fingertip blood glucose measurement. In the Figure 2, we incorporated subgroup and interaction analyses of VMBG, FMBG, and AKI. Venous blood glucose was less influenced by local factors and can more precisely reflect the actual blood glucose level within the body. The subgroup analysis demonstrated that irrespective of whether the patient developed AKI, the venous blood glucose level was significantly associated with the risk of death, indicating a severe stress response and metabolic disorders. While FMBG was associated with the 30-day mortality rate in the AKI group, there was no apparent correlation in the non-AKI group. AKI was not merely a renal function impairment but also signified a systemic severe inflammatory response, endothelial damage, and microcirculation disorder[1]. At this juncture, the value of fingertip blood glucose not only represented the blood glucose concentration but also encompassed information regarding poor tissue perfusion and cellular metabolic disorder[2], thereby endowing it with a stronger prognostic signal. This instability also indirectly validated that the value of fingertip blood glucose as a predictor was not as significant as that of venous blood glucose. ("Results" section, line 388 and "Discussion" section, lines 523-533)

12.The interpretation of ROC analyses alongside survival models would be strengthened by reporting model calibration metrics, such as calibration plots or Brier scores.

We express our sincere gratitude for the constructive suggestion put forward by the reviewer. We documented the calibration metrics of the survival models for the original cohort and the Landmark cohort using the Brier score. This documentation was presented in Results section, lines 228-229, as well as in Supplementary Table 2 and Supplementary Table 4. The Brier scores of these models were all below 0.10, suggesting a high level of consistency between the predicted probabilities and the observed outcomes.

Table 1 The multiple severity-of-illness scores VIF values of the multivariate models of VMBG and FMBG

Multiple severity-of-illness scores VMBG model

VIF FMBG model

VIF

SOFA 2.55 2.50

SAPSⅡ 4.29 4.11

APSⅢ 3.88 3.80

OASIS 2.30 2.28

Table 2 The comparison between VMBG and FMBG within 30 days without trimming

Variables Overall 30-day survial 30-day mortality p

N 2699 2361 338

VMBG 125.14 [110.84, 147.47] 122.56 [109.50, 142.83] 146.56 [127.62, 175.77] <0.001

FMBG 133.60 [116.89, 156.40] 131.82 [115.50, 153.83] 150.66 [129.50, 174.64] <0.001

Table 3 Multivariate Cox regression analysis of VMBG, FMBG and 30-day mortality without trimming

Models HR(95%CI) P Brier

VMBG 1.010(1.001,1.012) �0.001 0.084

FMBG 1.008(1.005,1.011) �0.001 0.087

Adjusted for sex, age, race, comorbidity index, cerebrovascular disease, liver disease, chronic pulmonary disease, diabetes, congestive heart failure, cancer, renal disease, CRRT, ventilation, insulin, transfusion, SOFA, GCS, AKI stage, SAPSⅡ, APSⅢ, OASIS.

Table 4 The comparison between VMBG, FMBG and 30-day mortality of the landmark cohorts

Variables Overall 30-day survial 30-day mortality p

Landmark of 2 days’ point

N 2634 2332 302

VMBG 132.00 [113.33, 155.00] 130.00 [112.50, 153.00] 147.06 [124.58, 170.69] <0.001

FMBG 133.40 [114.00, 156.11] 131.50 [113.31, 154.00] 148.47 [124.42, 170.13] <0.001

Landmark of 3 days

N 2516 2242 274

VMBG 129.00 [113.00, 152.50] 127.67 [112.23, 150.00] 145.17 [125.04, 164.88] <0.001

FMBG 133.33 [115.00, 154.69] 131.61 [114.10, 152.89] 146.14 [124.88, 166.69] <0.001

Table 5 Multivariate Cox regression analysis of VMBG, FMBG and 30-day mortality of the landmark cohorts

Models HR(95%CI) P C-index Brier

Landmark at 2 days point

VMBG 1.004(1.001,1.008) 0.013 0.813 0.083

FMBG 1.004(1.001,1.008) 0.027 0.813 0.083

Landmark at 3 days point

VMBG 1.006(1.003,1.010) �0.001 0.812 0.080

FMBG 1.005(1.001,1.009) 0.021 0.811 0.080

Adjusted for sex, age, race, comorbidity index, cerebrovascular disease, liver disease, chronic pulmonary disease, diabetes, congestive heart failure, cancer, renal disease, CRRT, ventilation, insulin, transfusion, SOFA, GCS, AKI stage, SAPSⅡ, APSⅢ, OASIS.

Figure 1 Number of venous glucose measurements within 30 day (A). Number of fingertip glucose measurements within 30 day (B).

Figure 2 Forest plots for subgroup analyses of VMBG and FMBG with 30-day mortality

References

[1] Lee SA, Cozzi M, Bush EL, Rabb H. Distant Organ Dysfunction in Acute Kidney Injury: A Review. Am J Kidney Dis. 2018;72(6):846-856. doi: 10.1053/j.ajkd.2018.03.028. PMID: 29866457.

[2] Kanji S, Buffie J, Hutton B, Bunting PS, Singh A, McDonald K, et al. Reliability of point-of-care testing for glucose measurement in critically ill adults. Crit Care Med. 2005;33(12):2778-85. doi: 10.1097/01.ccm.0000189939.10881.60. PMID: 16352960.

Once again, thank you very much for your comments and suggestions.

Yours sincerely,

Yun Liu

2026-02-02

---

## [Decision Letter · Decision Letter 3]

7 Feb 2026

Differential analysis of mean blood glucose levels from venous and fingertip in predicting 30-day mortality among ICU patients with severe trauma: A retrospective study utilizing the MIMIC-IV database

PONE-D-25-23041R3

Dear Dr. Liu,

We’re pleased to inform you that your manuscript has been judged scientifically suitable for publication and will be formally accepted for publication once it meets all outstanding technical requirements.

Kind regards,

Aleksandra Klisic

Academic Editor

PLOS One

Additional Editor Comments (optional):

Reviewers' comments:

Reviewer's Responses to Questions

**Comments to the Author**

Reviewer #3: All comments have been addressed

2. Is the manuscript technically sound, and do the data support the conclusions?

Reviewer #3: Yes

3. Has the statistical analysis been performed appropriately and rigorously?

Reviewer #3: Yes

4. Have the authors made all data underlying the findings in their manuscript fully available?

Reviewer #3: Yes

5. Is the manuscript presented in an intelligible fashion and written in standard English?

Reviewer #3: Yes

Reviewer #3: (No Response)

**Do you want your identity to be public for this peer review?** For information about this choice, including consent withdrawal, please see our Privacy Policy

Reviewer #3: **Yes:** Fateme Bnayi

---

## [Editor Report · Acceptance letter]

PONE-D-25-23041R3

PLOS One

Dear Dr. Liu,

I'm pleased to inform you that your manuscript has been deemed suitable for publication in PLOS One. Congratulations! Your manuscript is now being handed over to our production team.

Kind regards,

on behalf of

Dr. Aleksandra Klisic

Academic Editor

PLOS One